# Interacting evolutionary pressures drive mutation dynamics and health outcomes in aging blood

Kimberly Skead [1,2,3], Armande Ang Houle[1,2], Sagi Abelson[1,2], Mawusse Agbessi[1], Vanessa Bruat[1], Boxi Lin[1], David Soave[1,4], Liran Shlush[5], Stephen Wright [6], John Dick [1,2,7], Quaid Morris [2,3,8 ✉] & Philip Awadalla [1,2,9 ✉]

Age-related clonal hematopoiesis (ARCH) is characterized by age-associated accumulation of somatic mutations in hematopoietic stem cells (HSCs) or their pluripotent descendants. HSCs harboring driver mutations will be positively selected and cells carrying these mutations will rise in frequency. While ARCH is a known risk factor for blood malignancies, such as Acute Myeloid Leukemia (AML), why some people who harbor ARCH driver mutations do not progress to AML remains unclear. Here, we model the interaction of positive and negative selection in deeply sequenced blood samples from individuals who subsequently progressed to AML, compared to healthy controls, using deep learning and population genetics. Our modeling allows us to discriminate amongst evolutionary classes with high accuracy and captures signatures of purifying selection in most individuals. Purifying selection, acting on benign or mildly damaging passenger mutations, appears to play a critical role in preventing disease-predisposing clones from rising to dominance and is associated with longer disease-free survival. Through exploring a range of evolutionary models, we show how different classes of selection shape clonal dynamics and health outcomes thus enabling us to better identify individuals at a high risk of malignancy.

[1] Ontario Institute for Cancer Research, Toronto, ON, Canada. [2] Department of Molecular Genetics, University of Toronto, Toronto, ON, Canada. [3] Vector Institute for Artificial Intelligence, Toronto, ON, Canada. [4] Department of Mathematics, Wilfrid Laurier University, Waterloo, ON, Canada. [5] Department of Immunology, Weizmann Institute of Science, Rehovot, Israel. [6] Department of Ecology and Evolutionary Biology, University of Toronto, Toronto, ON, Canada. [7] Princess Margaret Cancer Centre, Toronto, ON, Canada. [8] Computational and Systems Biology Program, Memorial Sloan Kettering Cancer Center, New York, NY, United States. [9] Dalla Lana School of Public Health, University of Toronto, Toronto, ON, Canada. ✉email: morrisq@mskcc.org; Philip.awadalla@oicr.on.ca

Hematopoiesis proceeds through an extensive differentiation hierarchy rooted in a population of hematopoietic stem cells (HSCs)[1]. The HSC pool is estimated to consist of between 10,000 to 200,000 cells[2,3] and is among the most productive and tightly regulated populations in the human body. As individuals age, somatic mutations accumulate in HSCs, or in early blood cell progenitors[3–6]. Some mutations confer a proliferative advantage to certain cells, clones, in the hematopoietic hierarchy and result in a disproportionate lineage representation in the mature blood cell pool[3–6]. This predicted imbalance is observed with increasing frequency as individuals age and accordingly has been called Age-Related Clonal Hematopoiesis (ARCH) or, alternatively, Clonal Hematopoiesis of Indeterminate Potential (CHIP)[3–7]. ARCH has been linked to an increased risk for cancer and various cardiovascular (CVD) conditions including inflammation, atherosclerosis, thrombosis, and sudden death[3–7]. However, only a small proportion of individuals with ARCH progress to disease, and mechanisms driving the transformation to malignancy remain unclear. With respect to cancer, specifically Acute Myeloid Leukemia (AML), the presence of mutations in known AML driver genes at a high frequency is one of the best predictors of later disease onset. However, these same mutations are observed in healthy individuals who display no signs of hematological malignancy[7].

The evolutionary trajectory of somatic mutations in cellular populations is governed by a combination of deterministic processes, selection, and stochastic neutral processes (genetic drift)[8]. Mutational profiles from blood afford us a unique opportunity to study somatic evolutionary processes within, and among, individuals. Each blood sample is a reflection of the population history of the aging HSC pool, as well as the derived cell populations. Through high-coverage sequencing, we are able to capture the full spectrum of mutational variation within each blood population, including variants segregating at extremely low frequencies which are likely to be the targets of negative selection, and which are typically not captured in low-coverage to moderate-coverage sequencing efforts.

The ability to detect and quantify negative selection would allow us to move beyond the comparison of exclusively adaptive versus neutral models, which are conventionally used to model cancer evolution[9–11], and explore more diverse models that consider negative selection[12,13]. For example, the genetic background on which driver mutations arise has not been well characterized and could explain variation in outcomes. Indeed, little attention has been paid to the role of mutations occurring in non-driver genes in shaping disease outcomes[14]. For the remainder of this paper, we will refer to mutations accumulating in non-driver genes as "passenger mutations". Many passenger mutations are neutral and will have no impact on the fitness of the clone, however, some passenger mutations will be mildly damaging and, should they occur within a clone carrying a driver mutation, could impact the rate of expansion of the driver-harboring clones (Fig. 1a)[15]. As the majority of mutations (>99.9%) in cancers are passenger mutations, modeling their appearance and contribution to clonal fitness is critical to understanding mutation rates, cellular evolution, and disease progression[16]. As such, to fully exploit the predictive potential of ARCH for cancer and CVD outcomes, it is critical that we consider the full range of selective events occurring in the mature blood pool.

Here, we use advanced statistical and deep learning techniques to study the underlying evolutionary mechanisms driving cellular dynamics in pre-cancerous and normal hematopoietic populations. We generated somatic variant calls from 92 individuals who subsequently progressed to AML (preleukemic cases), and 385 age-matched and sex-matched healthy controls from the European Prospective Investigation into Cancer and Nutrition (EPIC) study[7,17]. Error corrected sequencing was performed on whole blood for 261 genes (xGen® AML Cancer Panel) implicated in AML at approximately 5000× coverage and is described in detail elsewhere[7].

Our approach to estimating intra-evolutionary processes underscoring hematopoietic dynamics is outlined in Fig. 1b–d. Briefly, we consider sequencing data from each individual to be derivative of a hematopoietic stem cell population and extract population-level summary statistics to describe patterns of somatic mutations in the mature blood cell pool (Fig. 1b). Summary statistics include counts of mutations, both overall and parsed according to whether they are silent or missense/nonsense, as well as summaries of variant allele frequency[18,19]. Summary statistics, such as these and others, are frequently used to test for departures from neutrality in population genetics and, collectively, can be used to discriminate among mutation rates and selective pressures acting on polymorphisms segregating in genomic regions of interest[10,11,18–22]. Using a deep learning classifier, trained and tested using clonal genomic sequences simulated across a range of realistic evolutionary scenarios, we demonstrate that we are able to distinguish between different evolutionary pressures in populations with high accuracy. Subsequently, we use our trained classifier to show how mutation rate and selection work in conjunction to influence the evolutionary trajectories of mutations in health and preleukemic blood cell populations and highlight the critical role that mildly damaging mutations play in preventing cancer prevention.

## Results

**Multi-task deep neural networks recover evolutionary dynamics and parameters with high accuracy.** To infer evolutionary processes acting within each blood cell population, we trained an ensemble of deep neural networks (DNNs), hereafter "classifier", using summary statistics derived from populations simulated across a range of evolutionary scenarios as input features, (Fig. 1b–d). A total of 4.6 million simulations were produced and we used these to create a look-up grid of parameter combinations representing a comprehensive range of plausible evolutionary scenarios (see "Methods" section, Table 1, Supplementary Fig. 1). Forward simulations were performed and four parameters were varied between simulations: mutation rate, probability of a mutation being beneficial ($p$), coefficient of positive selection ($sp$; corresponding to the relative fitness advantage of cells with this mutation), and coefficient of negative selection ($sn$; relative fitness disadvantage of cells with this mutation)[23]. In our simulations, we expect selection to act on nonsynonymous sites and synonymous sites were simulated as evolving under a neutral model. Each unique combination of the four parameters corresponds to a distinct evolutionary model. However, each model can be collapsed into one of four overarching evolutionary classes: neutral (no selection), positive selection only, negative selection only, and combination models which allow for the accumulation of mutations subject to both positive and negative selection within the same cellular population (see "Methods" section). Each neural network within our ensemble classifies a population into one of the four evolutionary classes and estimates the four parameters comprising a given model (Fig. 1c). Through comparing the outputs of our classifier, we can determine the uncertainty in the classifications (Fig. 1d)[24].

We were able to obtain a high classification accuracy for populations simulated under positive-only (0.99) and combination (0.97) models and relatively high accuracy for populations simulated under neutral (0.80) and negative-only (0.83) models when testing our classifier on a held-out test set of simulations (10% of our simulated data) (Fig. 1e, Supplementary Fig. 2). A

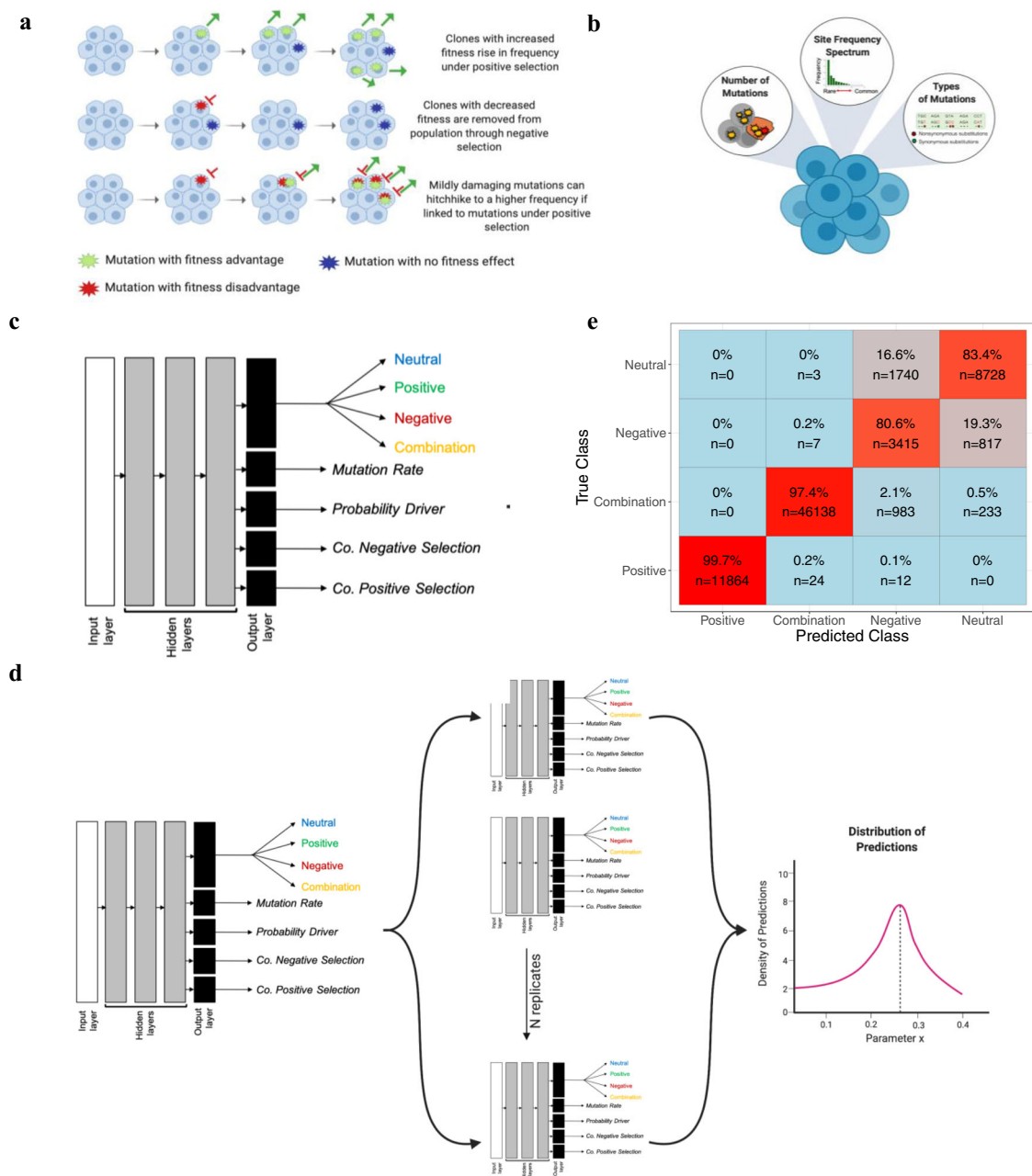

**Fig. 1 Deep learning models can be used to discriminate amongst the evolutionary pressures shaping blood evolution. a** The impact of selection and genetic drift on shaping clonal dynamics. Cells accumulate somatic mutations in each division. The majority of mutations will be either neutral (blue) or mildly damaging (red). Driver mutations will increase the fitness of a cell and increase the frequency in the population (green). However, mutations are also able to rise in frequency through genetic drift. **b** Mutation summary statistics extracted from blood cell populations. Summary statistics fall into three categories: (1) Counts of mutations in each blood sample (overall and stratified according to mutation type (silent and missense) across variant allele frequency intervals, (2) The frequency of mutations (variant allele frequency), and (3) mutation annotation and respective ratios (proportion of missense relative to total missense sites over the proportion of silent mutation relative to total silent sites). A total of 16 summary statistics are extracted from each population. **c** Deep Neural Network Architecture. Each DNN was trained as a multi-task neural network and classifies a population into one of four overarching evolutionary classes and predicts four continuous parameters. Each neural network consisted of an input layer (16 units with each unit corresponding to a summary statistic), three hidden layers (512 units), and five output layers which included the classification output (four units) and four regression outputs (one unit each). **d** DNN Ensemble. We trained a total of ten deep neural networks (DNNs) independently, yet with identical architecture. Through employing an ensemble-based approach, we are able to obtain a distribution of predictions for each population. **e** Classification performance for simulated evolutionary classes. The *y*-axis represents the true evolutionary class, and the *x*-axis represents the predicted evolutionary class. Classification accuracy ranges from blue (low accuracy) to red (high accuracy). We obtain a high classification accuracy across evolutionary classes (94.8%). Positive and combination classes are predicted with 99.7% and 97.4%, respectively. We observe a reduction in accuracy in neutral (80.6%) and negative (83.4%) classes of evolution.

**Table 1 Parameter ranges used to generate evolutionary simulations.**

| Class | Mutation rate ($\mu$) | Probability of a mutation being a driver mutation ($p$) | Coefficient of positive selection ($s_p$) | Coefficient of negative selection ($s_n$) |
|---|---|---|---|---|
| Neutral | All | 0 | 0 | 0 |
| Positive selection | All | $0 < p < 0.5$; $p = 1$ | $0 < s_p < 0.05$ | 0 |
| Negative selection | All | 0 | 0 | $0 < s_n < 0.05$ |
| Combination | All | $0 < p < 1$ | $0 < s_p < 0.05$ | $0 < s_n < 0.05$ |

perturbation analysis of our inverse model showed that we can predict positive-only or combination evolutionary classes with high certainty but that our model has some difficulty distinguishing between neutral and negative-only selection when there were few mutations. To ensure that our model was able to perform well on data from evolutionary parameters not included in our training data, we generated a novel set of simulations using new parameter combinations that do not appear in the training set. We find that we are able to achieve a similar degree of accuracy in evolutionary class prediction (Supplementary Fig. 3). Fewer mutations, or a lower level of variability in the population, could arise following a selective sweep or when populations are subject to a lower mutation rate (Supplementary Fig. 4). In addition, a lack of mutational information could be attributable to the selective removal of SNPs as a result of negative selection (Supplementary Fig. 5). Distinguishing between neutral evolution and negative selection is a challenge in population genetics as weakly damaging mutations can segregate in the population at low frequencies and have a mild impact on reducing variability at linked loci[12,13]. Further, while we can distinguish between overarching evolutionary classes with high accuracy, as well as the presence or absence of positive or negative selection, our model struggles to discriminate amongst the weaker coefficients of selection which is notoriously challenging in population genetics[12]. As such, we limit our inferences of selective dynamics in blood to the overarching evolutionary class which we are able to discriminate with high accuracy.

**Hematopoietic populations show evidence of positive and negative selection regardless of disease outcome.** We applied our classifier to preleukemic cases and healthy controls to infer population-level evolutionary processes. We find that in the majority of individuals (71%), both controls and cases, the hematopoietic population does not evolve neutrally (Fig. 2a). We reject models of neutral evolution in the majority of cases (79%) and controls (64%) and we observe a significantly higher departure from neutrality in cases than controls ($\chi^2(1) = 7.32$, $p$-value $= 0.007$, $n_{cases} = 73$, $n_{control} = 246$). The majority of cases (62%) and the plurality of (43%) control fit combination classes of evolution; in other words, we are able to detect signatures of both positive and negative selection in their blood biopsy indicating a functional impact of negative selection acting on passenger mutations. We observe that higher levels of predictive uncertainty correspond with a reduction in segregating mutations in the mature blood cell pool, in keeping with our classifier's performance on simulated populations (Supplementary Fig. 6). In summary, few cases or controls are evolving neutrally, and we find evidence of positive selection and of negative selection in the majority of preleukemic cases and healthy controls.

**Hematopoietic populations evolve in an age-dependent manner.** As ARCH is known to be an age-associated phenomenon, we investigated if there is any association between the age of an individual and the selective pressures governing their hematopoietic dynamics. Participants were binned into age groups

spanning ten-year intervals and the proportion of participants in each age range fitting each evolutionary class was calculated. We find clear associations with age and the dominant class of evolution in individuals (Fig. 2b). Specifically, we observe an age-related decline in the proportion of controls fitting negative-only or neutral classes of evolution and a parallel increase in controls fitting the combination class. Our results are consistent with a model in which individuals accumulate passenger mutations as they age, some of which will have a slightly damaging effect. In parallel, with increasing mutation accumulation, there is an increased likelihood of a rare driver event occurring which would cause an individual to shift to a combination, or positive-only, class of evolution. We find that many preleukemic cases show evidence of positive selection at a younger age than controls. In particular, in preleukemic cases, where age-associated clonal expansions have been previously reported, we observe an increased overall proportion of individuals fitting combination models in younger age groups indicating that driver events have occurred earlier. At a young age, driver mutations are likely to be arising on a background with fewer mildly damaging passenger mutations and thus may experience a relatively higher fitness advantage compared to the same mutation arising on a background with a greater number of mildly damaging passenger mutations; a hypothesis that we investigate in the following section.

**Controls have a higher proportion of passenger-to-driver mutations than cases.** To investigate if the proportion of mutations in known driver genes compared to non-driver genes could explain some variation in outcomes, we compared the types and patterns of mutations between the cases and controls. We annotated mutations as drivers if they occurred in driver genes found to be highly mutated in the Cancer Genome Atlas Acute Myeloid Leukemia project (Supplementary Fig. 7). We first asked whether cases simply have more mutations, thus predisposing their blood populations to cancer. Consistent with the previous reports[7], our cases had more mutations on average than age-matched healthy controls (Wilcoxon rank-sum test, $W = 12,196$, df $= 1$, $p$-value $= 2.9\mathrm{e}{-06}$, $n_{cases} = 92$, $n_{controls} = 385$). and, in the combination class of evolution, more mutations in known driver genes (Supplementary Fig. 7). A higher mutation count in cases is consistent with our classifier's prediction that there is a small increase in mutation rate in a preleukemic context (mean mutation rate: $\mu = 1.2\mathrm{e}{-10}$ per bp per division) compared to healthy controls (mean mutation rate: $\mu = 1.1\mathrm{e}{-10}$ per bp per division) (Wilcoxon rank-sum test, $W = 14336$, df $= 1$, $p$-value $= 0.004$, $n_{cases} = 92$, $n_{controls} = 385$) (Fig. 2c–d). We estimated the mutation rate assuming a population size of 10,000 and we have scaled our estimates to account for varying estimates of HSC population size (Supplementary Figs. 8–10).

Intriguingly, we do observe mutations in driver genes in healthy controls fitting positive models of selection. It is possible that these individuals do not progress to disease if driver mutations are arising in competing clones, thus preventing one clone from rising to dominance. However, another possible

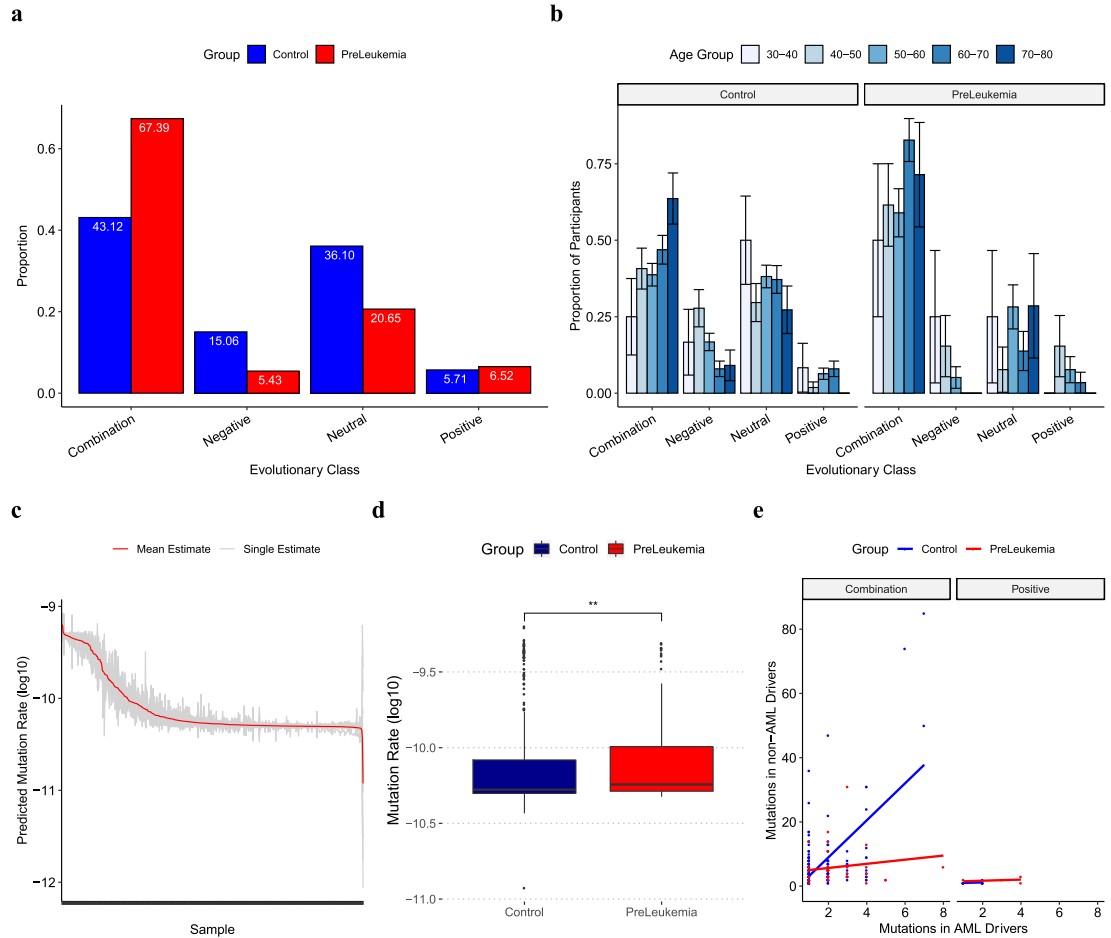

**Fig. 2 Hematopoietic evolution is governed by a range of evolutionary dynamics. a** Evolutionary classes in preleukemic (red) and healthy (blue) blood populations. The majority of blood populations do not evolve neutrally (72%). Similarly, only 9% of individuals fit positive models of evolution. Populations do not evolve neutrally in the majority of preleukemic cases (79%) and healthy controls (64%). The majority of preleukemic (62%) and the plurality of healthy (43%) individuals fit combination (both beneficial and damaging mutations arising) classes of evolution. **b** Age-associations across evolutionary class predictions. Participants were stratified into 10-year age windows. Age intervals range from 30–40 (light blue) to 70–80 (dark blue). Each bar represents the proportion of individuals within each age bin fitting each evolutionary class for preleukemic individuals ($n = 92$) and healthy controls ($n = 385$). Standard errors for each proportion were calculated by $p(1−p)/n$ where $p$ is the proportion of individuals fitting a particular class and $n$ is the total population. We observe significant differences in the proportion of individuals fitting combination classes of evolution in the 50–60 age range (Pearson's chi-squared test, $X^2 = 4.54$, p-value = 0.03), the 60–70 age range (Pearson's chi-squared test, $X^2 = 10.55$, p-value = 0.001), and in the neutral class of evolution in the 60–70 age range (Pearson's chi-squared test, $X^2 = 4.73$, p-value = 0.03). **c** Range of mutation rate estimations across a cohort of participants. We show the estimated mutation rate for each sample from each DNN in our ensemble (gray). The mean estimate from the classifier outputs is shown in red and samples are sorted by the mean estimated mutation rate. Mutation rates (y-axis) are log-transformed and scaled to a population size of 10,000. **d** Preleukemic blood populations have a higher mutation rate than healthy controls. Each boxplot illustrates the distribution of estimated mutation rates across samples grouped according to outcome status (control ($n = 385$): blue, preleukemic ($n = 92$) = red), the midline represents the medians, the upper and lower bounds the interquartile ranges, and the whiskers extend to 1.5 times the interquartile range. The level of significance is indicated as follows: ns: $p > 0.0$, *p-value <= 0.05, **p-value <= 0.01, ***p-value <= 0.001, ****p-value <= 0.0001. Preleukemic cases are found to have a modest yet significantly higher mutation rate than controls (Two-sided Wilcoxon rank-sum test, $W = 14336$, df = 1, p-value = 0.004). **e** Relative passenger to driver mutation proportion across evolutionary classes. The number of mutations in known driver genes is plotted against the number of mutations in non-driver genes for each individual blood population with healthy controls shown in blue and preleukemic individuals in red. We used linear regression to compare the relationship between the number of mutations falling into known driver genes versus non-driver genes in cases (red) and controls (blue) fitting combination and positive evolutionary classes. The 95% confidence level interval for predictions from each linear model is indicated in gray. In the combination model, we find a significant interaction between the number of mutations occurring in non-driver genes compared to driver genes in controls ($\beta = 5.76$) and cases ($\beta = 0.642$); $F (1, 224) = 28.5$, p-value = 2.23e−07. However, in the positive class, we did not find a significant interaction between the number of mutations occurring in non-driver genes compared to a driver in controls ($\beta = 0.16$) and cases ($\beta = 0.17$); $F(1,12) = 0.0004$, p-value = 0.98.

explanation for the differences in outcome is the proportion of driver to passenger mutations. Using linear regression, we compare the relationship between the number of mutations falling into known driver genes versus non-driver genes for cases and controls fitting the combination and positive evolutionary classes. In the combination model, we find a significant interaction between the number of mutations occurring in non-driver genes compared to driver genes in controls ($\beta = 5.76$) and cases ($\beta = 0.642$); $F (1, 224) = 28.5$, p-value = 2.23e−07. However, in the positive class, we did not find a significant interaction

between the number of mutations occurring in non-driver genes compared to driver genes in controls ($\beta = 0.16$) and cases ($\beta = 0.17$); $F(1,12) = 0.0004$, $p$-value $= 0.98$. However, in the positive comparison, our sample size is low, so we may not be powered to detect such a difference. The increased proportion of mutations in passenger genes compared to driver genes in the combination class is consistent with a model in which negative selection acting on mildly damaging passenger mutations is playing a protective role in inhibiting or stalling clonal expansions.

**Distinct patterns of inferred pathogenicity associate with evolutionary classes.** To determine whether some passengers are playing a protective role, we scored each mutation according to how likely it was to affect protein function and conservation after blood sample classification. In doing so, we can independently evaluate the performance of our evolutionary predictions. We scored mutations using the Combined Annotation-Dependent Depletion (CADD v. 1.4) algorithm (Fig. 3a)[25]. Using a combination of functional prediction, conservation, epigenetic measurements, gene annotations, and the sequence surrounding a given variant, CADD provides a measure of the functional impact of single nucleotide variants, and small insertions/deletions, in the genome. CADD scores assess whether a mutation alters protein function, and have difficulty distinguishing between whether it changes protein expression, inhibits its activity, or causes the protein to be constitutively active; as such we will call mutations with high CADD scores "function-altering".

Overall, we observe that mutations falling in known driver genes tend to have a higher CADD score than mutations in non-driver genes. However, in keeping with our expectations of neutral evolution, we do not observe a significant difference in CADD score between mutations in known driver genes ($n = 32$) and non-driver genes in neutral cases ($n = 899$) (Wilcoxon rank-sum test, $W = 7598.5$, $p$-value $= 0.3$), suggesting that these mutations are not function-altering and confer no relative fitness advantage or disadvantage to the clone in which they are found. In comparison, in individuals showing evidence of positive selection (positive ($n = 47$) and combination ($n = 401$), mutations in known driver genes had significantly higher CADD scores than mutations in non-driver genes ($n = 21$ and $n = 1487$, for positive and combination classes, respectively) (Wilcoxon rank-sum test, positive models: $W = 548$, $p$-value $= 0.001$; combination models: $W = 278,136$, $p$-value $< 2.2e{-}16$). Further, we observe that the average CADD score assigned to passenger mutations in negative-only models ($n = 153$) is significantly lower (Wilcoxon rank-sum test, $W = 39,298$, $p$-value $= 0.004$) than passenger mutation CADD scores in the neutral class ($n = 899$) suggesting that negative selection plays a role in removing the more damaging mutations and decreasing the overall pathogenicity of segregating mutations. The role of negative selection in decreasing the overall pathogenicity of the blood pool is further supported by the average CADD scores of passenger mutations in the combination class being greater and smaller than the average score of passenger mutations in the negative-only and neutral class, respectively. In the absence of recombination, mutations which would typically be removed are able to continue to segregate in the blood population in the presence of positively selected driver mutations, and, accordingly, we observe higher average pathogenicity of passenger mutations in the combination class. Finally, it is worth noting that the passenger mutations in the positive-only class have significantly lower pathogenicity than those in the combination class. Passenger mutations with higher pathogenicity are likely to be subject to stronger negative selection thus conferring a protective effect to the individual in the presence of positive selection acting on drivers. A better understanding of how these potentially protective mutations are distributed across genes would allow us to identify which genes might be critical in preventing clonal expansions.

**Clusters of genes are enriched for function-altering mutations across evolutionary classes.** To investigate if certain genes are more frequently found to play a protective role when mutated, we determined which genes are enriched for function-altering mutations in each evolutionary class, as well as the overlap of genes with function-altering mutations across evolutionary classes (Fig. 3b). For this comparison, we used a lower threshold of a CADD score of 10 to determine which mutations are likely to be function-altering. The majority of genes harboring function-altering mutations are observed in combination and neutral classes of evolution. However, there are subsets of genes that are enriched exclusively for function-altering mutations in the presence of positive or negative selection. Reassuringly, we find that many known driver genes (*DNMT3A, TET2, IDH2, TP53*) are enriched for function-altering mutations among positive and combination classes of evolution only and not among neutral or negative classes of evolution (Supplementary Fig. 11). Further, we observe that there is an overlap of genes enriched for function-altering mutations in negative-only and combination classes of evolution indicating that these genes might experience stronger negative selection.

We next asked if the inferred pathogenicity of mutations in the dominant clone corresponds with the frequency at which it is observed in the mature blood cell pool. To do so, we evaluated the relationship between the CADD score and the frequency of the dominant clones, defined as the clone with the highest variant allele frequency in an individual, in each class (Fig. 3c). We find that clones are able to rise to fixation in the absence of both negative and positive selection where the primary driving force of evolution is genetic drift. Clones rising to a high frequency stochastically could, in part, be explained by a reduction in the effective population size of the HSC population owing to a small population of stem cells with a higher fitness dominating blood cell production. With a reduced population size, mutations are able to rise to a higher frequency and become fixed in a population more rapidly. However, only mutations with a low CADD score are found at high frequencies in the neutral class. As expected, in the presence of negative selection, we observe a depletion of clones in the higher pathogenicity categories as they have likely been removed by selection. Clones that persist in the negative-only model could indicate a functional threshold at which mutations are not efficiently removed by selection and continue to segregate in the population. Conversely, in the positive-only class, clones, including those with high pathogenicity, are found at higher frequencies. We observe a higher variance in CADD scores in the combination class which is consistent with our expectation that, when neither positive nor negative selection are able to act efficiently, variants will segregate at intermediate frequencies rather than sweeping to high fixation or being purged from the population, respectively.

**Selective interference may be associated with slowing clonal expansions.** Having established that in the combination class, controls have significantly higher non-driver to driver ratios and that these non-drivers have significantly higher CADD scores than those in the positive-only class, we then ask whether non-drivers played a role in preventing progression to AML through selective interference. Selective interference is particularly relevant as studies report that driver mutations, while found in both healthy controls and preleukemic cases, tend to segregate at a much higher frequency in a preleukemic context[7]. We propose

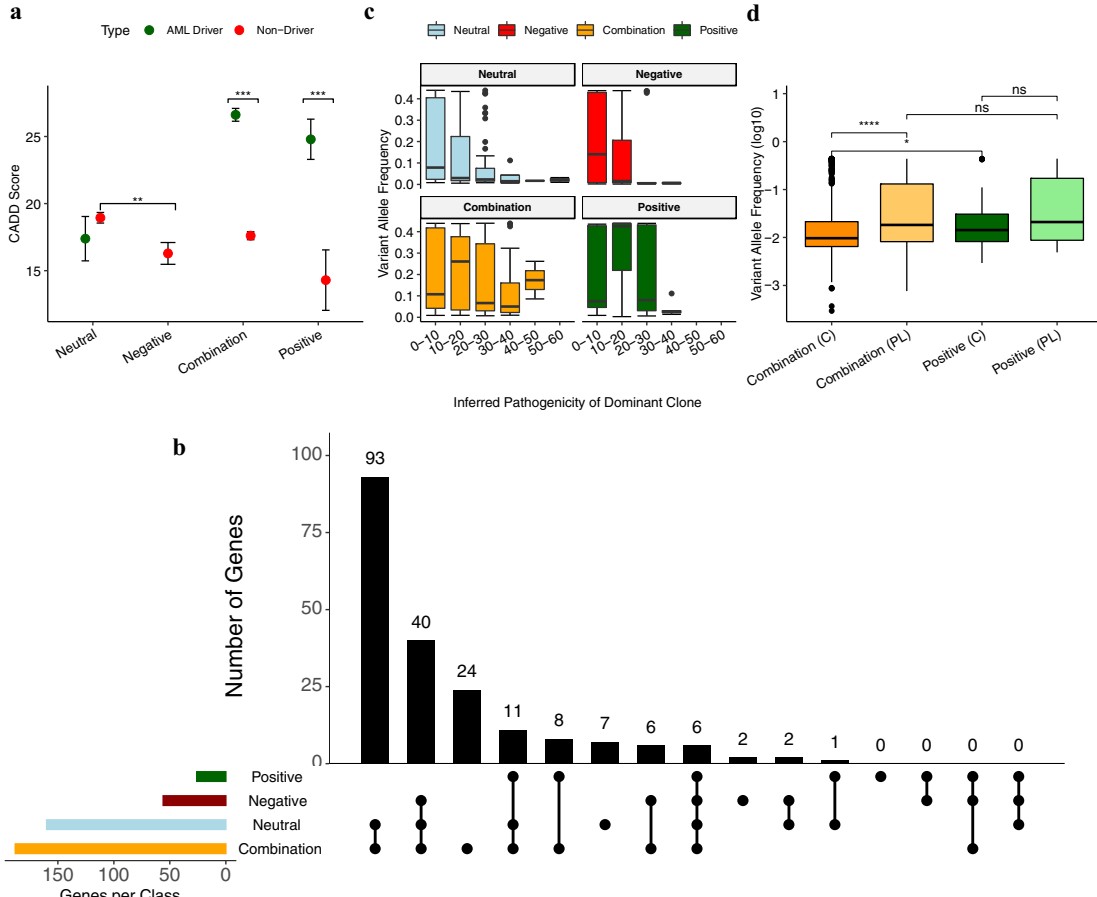

**Fig. 3 Distinct patterns of inferred pathogenicity and clonal dynamics are associated with evolutionary class predictions. a** Predicted functionality of mutations in each evolutionary class. Average CADD scores were calculated for mutations in known driver genes (green) and non-driver genes (red) and are presented as mean values $+/-$ SEM. The level of significance is indicated as follows: ns: $p > 0.0$, $^*p$-value $<= 0.05$, $^{**}p$-value $<= 0.01$, $^{***}p$-value $<= 0.001$, $^{****}p$-value $<= 0.0001$. We capture a significant (Two-sided Wilcoxon rank-sum test, positive models: $W = 548$, $p$-value $= 0.001$; combination models: $W = 278136$, $p$-value $< 2.2e-16$) enrichment of high CADD scores in driver genes compared to non-driver genes. We do not observe a significant difference between CADD scores across mutations in driver genes and non-driver genes in neutral classes (Two-sided Wilcoxon rank-sum test, $W = 7598.5$, $p$-value $= 0.3$), The average CADD score assigned to passenger mutations in negative models is significantly lower (Two-sided Wilcoxon rank-sum test, $W = 39298$, $p$-value $= 0.004$) than passenger mutation CADD scores in the neutral class. **b** Distribution of function-altering mutations in genes across evolutionary classes. The UpSet plot shows the distribution of function-altering mutations (CADD > 10) falling in genes across patients in different evolutionary classes. The total number of genes mutated in each evolutionary class is shown on the left (positive = green, negative = red, combination = orange, neutral = blue). The dark circles indicated classes with overlapping genes and the connecting bar indicated multiple overlapping genes. **c** Inferred pathogenicity of the dominant clones is correlated with the variant allele frequency across different evolutionary classes. We isolated the dominant clone within each individual blood pool. The CADD scores of each clone were binned into intervals of 10 and each boxplot illustrates the distribution of variant allele frequencies for each interval, the midline represents the medians, the upper and lower bounds the interquartile ranges, and the whiskers extend to 1.5 times the interquartile range. Plots are faceted according to evolutionary class (positive ($n = 30$): green, negative ($n = 65$): red, combination ($n = 229$): orange, neutral ($n = 158$): blue). We observe a wider distribution of CADD scores in neutral and combination models. In positive classes of evolution, clones are found at a higher frequency suggesting that they sweep to fixation. Similarly, in negative models of evolution, we observe a surplus of clones in the low CADD score bins which appear to segregate at a lower frequency. **d** Impact of Negative Selection on Clonal Expansions. We investigated if negative selection acting on passenger mutations impacted clonal expansions in a healthy and preleukemic context. To do so, we plotted the log-transformed variant allele frequency (VAF) of mutations found in cases and controls predicted to be evolving in positive class (green) and combination (orange) classes. Each boxplot illustrates the distribution of log10-transformed VAFs for each group, the midline represents the medians, the upper and lower bounds the interquartile ranges, and the whiskers extend to 1.5 times the interquartile range. The evolutionary class is denoted on the $x$-axis (combination/positive) and preleukemic/control status is indicated by a (PL) or (C), respectively. The level of significance is indicated as described above. VAF distributions were plotted separately for preleukemic individuals (light green/ light orange) and healthy controls (dark green/dark orange). We find that we are not able to discriminate between VAF distributions of mutations in healthy and preleukemic individuals in the positive class. Further, we are not able to discriminate between positive models of evolution and preleukemic individuals who fit combination models of evolution. However, we find that clones in controls fitting combination classes of evolution have a significantly lower VAF distribution compared to both preleukemic cases fitting combination models (Two-sided Wilcoxon rank-sum test, $W = 254988$, $p$-value $< 2.2e-16$, $n_{combination(PL)} = 403$, $n_{combination(C)} = 1095$) and clones in controls fitting positive models (Two-sided Wilcoxon rank-sum test, $W = 25180$, $p$-value $= 0.02$, $n_{positive(c)} = 34$).

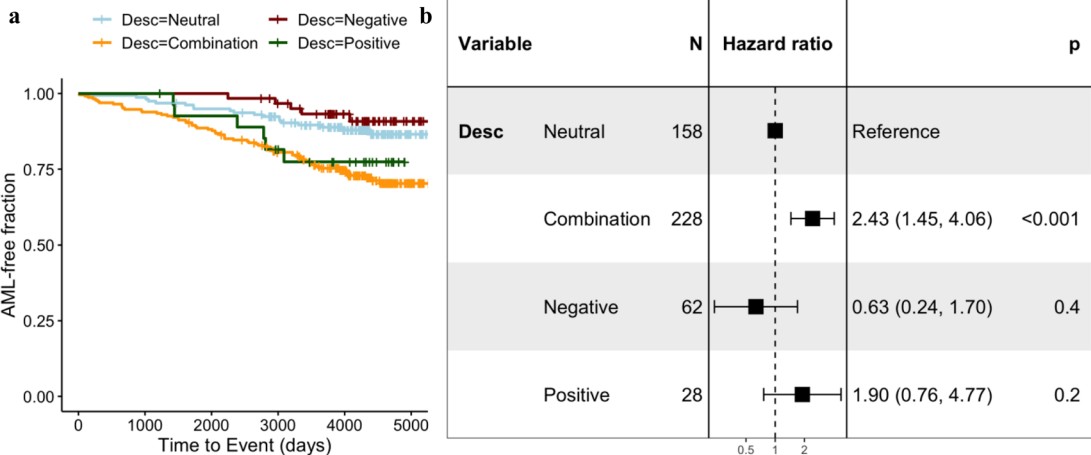

**Fig. 4 Negative selection is associated with AML-free survival. a** Kaplan–Meier curves of AML-free survival among EPIC participants. Survival is defined as the time between blood sample collection and diagnosis (for cases), for control survival is right-censored at last follow-up. Survival curves are stratified according to evolutionary class (positive = green, negative = red, combination = orange, neutral = blue). **b** Forest Plot of risk of AML development across evolutionary classes. Each row indicates hazard ratios associated with each evolutionary class (reference = neutral class). The horizontal lines indicate a 95% confidence interval for each class.

that selective interference, where the linkage between sites under multiple selective pressures will define the overall impact of selection acting on the population, could play a role in preventing mutations from rising to a high frequency in controls either through passenger mutations hitchhiking within the same clones as driver mutations, or if driver mutations arise in different clones and are competing for dominance in a finite cell pool[13,26,27]. We expect that clones under purely positive selection will be found at higher frequencies in blood compared to those which are subject to a combination of positive and negative selection where interference might play a role in preventing selective sweeps.

We find that mutations in preleukemic cases fitting a combination class ($n = 403$) tend to segregate at a significantly higher frequency compared to controls ($n = 1095$) (Wilcoxon rank-sum test, $W = 254,988$, $p$-value $< 2.2e-16$) (Fig. 3d). However, we do not observe a difference in the frequency at which mutations are found to segregate between cases and controls fitting positive models of evolution or between cases fitting combination and positive models of evolution. Decreased variant allele frequencies in healthy individuals fitting combination models are consistent with our prediction that selection acting on a subset of mutations in healthy controls prevents progression to disease even in the presence of positive selection. However, we do observe signatures of negative selection in preleukemic contexts which suggests that the impact of selection acting on passenger mutations, while detectable through our methods which incorporate multiple summaries of the data, remains negligible with respect to clonal progression. Further, we do observe that, while not significant owing to sample size, preleukemic cases fitting combination models tend to have a later age of diagnosis than preleukemic cases fitting positive-only evolutionary classes indicating that negative selection might play a role in slowing progression to disease.

Indeed, we find that ARCH occurring in the absence of positive selection, that is individuals who fit negative or neutral classes of evolution, is associated with a lower risk of progression to AML compared to individuals who have signatures of positive selection (log-rank test, $p$-value $= 2e-04$) (Fig. 4a). We find that individuals who fit combination classes of evolution have an approximately two-fold increased risk of progressing to AML compared to individuals fitting neutral models of evolution (hazard ratio, 2.43; 95% confidence interval, 1.45–4.06, Fig. 4b).

Owing to the small number of individuals fitting positive only classes of evolution, we cannot infer if the negative selection acting on passenger mutations in individuals fitting combination classes of evolution reduces the risk of progressing to AML. However, a scenario where multiple clones compete for dominance, thus maintaining clones at intermediate frequencies, would explain the greater risk conferred to individuals fitting combination classes of evolution. Our findings suggest that not all passenger mutations are equal in that some might be more efficient in preventing disease-associated clonal expansions. Further, through accounting for mutations that segregate alongside driver mutations, we would be able to greatly improve our understanding of ARCH as a biomarker for disease and better predict who is at risk of progressing to cancer.

## Discussion
Our ability to determine the evolutionary processes governing the impact of age-associated mutations on hematopoietic fitness is dependent on the deep genomic interrogation of blood profiles and our ability to discriminate amongst alternative evolutionary models. Estimates of cellular evolutionary features are dependent on the simultaneous inference of mutation and selection. Most tests to detect selection are based on the null hypothesis that mutant alleles are selectively neutral and have proven to be highly successful in detecting positive selection in germline and somatic tissues. However, the detection of negative selection in populations has remained challenging as they are reliant on the capture of rare variants which, prior to deep coverage sequencing, are typically not reliably detected in cellular populations[12]. Many tests of neutrality, including $dN/dS$ ratios which are often used to study selection in paired tumor-normal samples, are only sensitive to a conservative range of selection parameters and can be confounded by the timing of events. However, determining the mode of selection acting on a mutation is a key parameter to understanding tumor evolution as it offers critical insight into the evolutionary fate of the mutation in the population. In a somatic context, this becomes increasingly more important as negative selection acting on mutations co-occurring in the same clone as positively selected driver mutations could significantly alter the evolutionary trajectory of the mutant expected under positive or neutral evolution alone.

Using newly developed population-genetic neural network approaches which exploit the impact of linkage in clonal systems and combinations of informative summaries of mutation data[12,13], we are able to discriminate signatures of negative selection from neutrality. Our methods can classify different selection scenarios in cells sampled from individual liquid biopsies and enable us to evaluate the combined impact of both positive and negative mutations on patterns of clonal dominance in the mature blood cell pool of healthy and precancerous individuals. We find that hematopoietic populations largely do not evolve neutrally and that the presence of negative selection acting on mutations in non-driver genes plays an important role in disease development across aging blood systems. In line with other studies, we find that we do observe ARCH occurring in blood populations presumed to be evolving strictly under neutral evolution, however, in most instances, the trajectory of clonal mutations in the blood appears to be governed by the complex interplay of both positive and negative selection[28]. Further, we estimate the rate at which mutations accumulate in the aging hematopoietic system in both healthy and premalignant contexts. The rate at which ARCH-associated mutations occur in healthy and premalignant contexts has, to our knowledge, not been previously characterized owing to genomic and statistical challenges.

Our approaches untangle interacting and sometimes confounding factors in somatic evolution. We captured an important phenomenon within blood: that damaging mutations accrue along with ARCH-associated driver mutations. Negative selection acting on linked passenger mutations might slow the rate of clonal expansions. The presence of negative selection acting on passenger mutations in the presence of driver mutations offers a potential explanation for why some individuals who harbor driver mutations do not progress to disease. However, negative selection appears to play an important role in lowing the overall pathogenicity of the mature blood cell pool, thus maximizing the fitness of the individual; a finding which suggests that there could be an advantage to retaining mildly damaging mutations in cellular populations if they confer a protective effect in the presence of a driver mutation. Indeed, our survival analyses demonstrate that the evolutionary forces shaping ARCH are the same in that patterns of clonal hematopoiesis fitting negative or neutral classes of evolution are typically associated with a lower risk of progression to AML. Future work with long-read sequencing and longitudinal sampling is required to experimentally phase somatic mutations genome-wide, will help determine whether such mutations are occurring in the same or different clones thus allowing us to investigate the effects of both individual mutations, as well as the epistatic interactions between linked mutations and clonal trajectories.

## Methods

**Sequencing and data processing in EPIC cohort**. Participant selection, sequencing, and mutation calls have been described in full elsewhere[7]. Briefly, all DNA samples were obtained from individuals who enrolled in the EPIC study between 1993 and 1998 for individuals who provided a blood sample prior to progressing to AML ($n = 92$) and from individuals who did not develop any hematological malignancy during follow-up ($n = 385$). The median age at the time of recruitment into the cohort was 57 years old (range: 36–74.4 years old). The median time between sample collection and AML diagnosis or censorship was 6.3 years (range: 0.03 years–12.4 years) and 11.9 years (range: 3.36 years–14.9 years), respectively. Cases and controls were matched to an approximately 1:4.5 case/control ratio. Error corrected duplex sequencing was performed on all samples for 261 genes (xGen® AML Cancer Panel, https://www.idtdna.com/pages/products/next-generation-sequencing/targeted-sequencing/hybridization-capture/predesigned-panels/xgen-aml-panel) implicated in AML at approximately 5000X coverage. Variants were retained if they were supported by a minimum of five reads with a minimum of two reads in each direction. Following variant calling, mutations were annotated using ENSEMBL v.58, and VAGrENT for transcript and protein effects[29]. Annovar was used for additional functional annotation[30].

**Calculation of summary statistics**. Summary statistics were calculated per individual with sequencing data from each individual considered to be derivative of an HSC population. We calculated a total of 16 summary statistics for each hematopoietic population including population genetics statistics used to describe the site frequency spectrum (Tajima's D[18] and Fay and Wu's H[19]), counts and ratios of synonymous and nonsynonymous mutations both overall and for low (>0.1), intermediate (0.1–0.8), and high (>0.8) variant allele frequency windows, and the number of mutations in known driver genes.

**Simulations of hematopoietic populations across evolutionary scenarios**. Hematopoietic populations were simulated as clonal haploid populations evolving forward in time using the software SFScode[23]. We simulated haploid populations with an effective population size of 10,000 and a null recombination rate. Our initial modeling of the HSC population size ($n = 10,000$) was derived from the Abkowitz et al. (2002) estimates of the number of HSCs in mammals. However, following the publication of Lee-Six et al., which used an Approximate Bayesian approach to estimate that the HSC pool ($n = 40,000–200,000$), we adjusted our mutation rate predictions to account for a range of HSC population sizes (as shown in Supplementary Fig. 10). The expected number of mutations in a population ($\theta$) is a product of the population size $N$ and the per generation mutation rate ($\mu$): $\theta = 4N\mu$. Estimates of mutation rate can be extended to account for the range of population size estimates (10,000–200,000) that exist, and as they evolve in the future, for the hematopoietic stem cell population. Mutation rates were determined by evaluating the (1) length of the sequencing panel used, (2) an average number of mutations observed in the EPIC cohort, and 3) estimates of population size.

A 1 Mb region of the genome with a 1:2 ratio of synonymous and non-synonymous positions was simulated, where synonymous sites are not subject to selection. We simulated a grid of evolutionary scenarios across a plausible range of parameters (Table 1). Selection coefficients affecting each nonsynonymous mutation are sampled from a gamma distribution[31]. Owing to variation associated with cancer evolutionary parameters, we simulated ranges of mutation rates (from 1e−7 to 1e−5 mutations per base pair per generation), proportions of beneficial mutations (from 0 to 1; 0 being that all mutations are deleterious and 1 is that all mutations are beneficial), and rate parameters for the gamma distribution from which selection coefficients are sampled (from 0.001 to 0.005). The shape ($\alpha$) and rate ($\beta$) parameters of the gamma distribution were also varied. Effectively, in a subset of simulations, the rate parameter for the gamma distribution of negative selection coefficients ranges from 0.001 to 0.02475. We compute the means of selection coefficients based on the rates and shapes of the tested gamma distribution ($\mu = \alpha/\beta$). For each set of parameters, we perform 2000 replicates. In addition, we simulated populations evolving under neutral models of evolution (no selection). Summary statistics for simulated populations were calculated as described for the EPIC participants. The number of mutations subject to positive selection in each simulation is a function of the probability of a mutation being beneficial ($p$) and the number of nonsynonymous mutations. Each simulation was classified into one of four overarching evolutionary classes if the following conditions were met:

**Probability of a mutation being beneficial**. While the probabilities of a mutation being beneficial range from 0 to 1, we limited the simulations used in our training and testing data set to values ranging between 0 and 0.5 and 1 (only positive selection), as we did not think that a scenario in which more than half the mutations are driver mutations was feasible.

**Software for statistical analyses**. Following the simulation of data using SFSCode, all training, and testing of deep neural networks, analyses and figures were performed in the R statistical programming environment (v.4.0.4). Libraries that are required include plyr v.1.8.6[32], dplyr v.1.0.5[33], UpSetR v.1.4.0[34], Rtsne v.0.15[35], lsmeans v.2.3.0-0[36], forestmodel v.0.6.2[37], survival v. 3.2-10[38], survminer v.0.4.9[39], and keras v.2.4.0[40]. Figures were generated using ggplot2 v.3.3.3[41], ggpubr v.040[42], ggplotify v.0.0.7[43], and patchwork v.1.1.1[44]. All Wilcoxon rank-sum tests are two-sided unless otherwise specified. The reproducible code is available at https://github.com/kimskead/popgenArch[45].

**Evaluating fit between simulated summary statistics and summary statistics from EPIC individuals**. t-Distributed Stochastic Neighbor Embedding (t-SNE) was used to visualize the distribution of summary statistics for simulated and observed blood populations. A total of 5,000 sets of summary statistics were randomly sampled from each simulated evolutionary class for a total size of 20,000 simulated sets of summary statistics. All of the summary statistics from the EPIC cohort were included. Dimensionality reduction was completed in R using the package Rtsne.

**DNN ensemble**. We collated summary statistics from each simulation to create our main training dataset of populations evolving under different selection pressures. Our simulations were subdivided into training (90%) and test (10%) sets. To create a unique training set for each instance of our DNN ensemble, we further sub-sampled our training set into 10 sets consisting of 250,000 unique evolutionary models. Sampling was done with replacement and the number of models in each class was downsampled to match the minority class to avoid class imbalance. We

trained a total of ten deep neural networks (DNNs) independently using simulated data only. Each neural network consisted of an input layer (16 units), three hidden layers (512 units), and five output layers which included the classification output (four units) and four regression outputs (one unit each). We applied dropout regularization (0.1) to each hidden layer and used a real activation function. We applied a sparse categorical cross-entropy loss function to the classification task and mean square error was used to calculate loss associated with the parameter estimations. Each model was trained using Adam with a batch size of 500 for 20 epochs[46]. All hyperparameters were manually optimized. Each DNN was trained as a multi-task neural network to both classify a population into one of four over-arching evolutionary classes (positive, negative, combination or neutral) and predict four continuous parameters (mutation rate ($\mu$), the probability of a mutation being beneficial ($p$), coefficient of positive selection, and coefficient of negative selection). Our ensemble-based approach runs for approximately 4 h on a 2.3 GHz Quad-Core Intel Core i7 Processor. A benefit of our ensemble-based approach is that, for each blood cell population, each DNN emits a softmax probability distribution across the four overarching evolutionary classes. In a conventional classification task, the class with the highest probability will be selected as the best fit. However, as we are employing an ensemble-based approach, we can obtain a distribution of predictions for each population so as to measure the uncertainty associated with each prediction. After each DNN was trained, we assessed our ability to accurately predict the true evolutionary class using a held-out set of simulated data which was split from the training data prior to sampling training sets for each DNN. Any instance where there are identical sets of summary statistics in the training and test set were removed from the test set prior to evaluation in order to eliminate information leakage. To ensure that our classifier was able to generalize well to parameter combinations not included in the training data, we generated an additional test set of simulated data wherein the gamma distributions from which parameters are drawn were modified thus generating novel values and parameter combinations not included in the training and test set. Following training and testing using simulated data, each trained DNN in our ensemble was used to estimate the overarching evolutionary class and evolutionary parameters using summary statistics derived from healthy and preleukemic blood populations. For each individual, we calculated the mean and standard error for each softmax probability across the four evolutionary classes and accepted the class with the maximum softmax probability as the class of best fit. Similarly, for each regression task, we calculate the mean and standard error for each parameter across the predictions for each individual.

**Predicting best fit class and estimating parameters for EPIC individuals**. To obtain the best fit evolutionary class for each individual, we calculated the mean and standard error for each softmax probability across the four evolutionary classes and accepted the class with the maximum softmax probability as the class of best fit. To calculate age distributions of evolutionary classes, we bin participants into 10-year age windows. We calculate the proportion of individuals within each age bin fitting a particular evolutionary class for preleukemic individuals and healthy controls. Standard errors of each proportion were calculated by $\sqrt{p \times (1-p)/n}$ where $p$ is the proportion of individuals fitting a particular class and $n$ is the total population. We calculate the mean and standard error for parameter predictions to estimates for mutation rate and the probability of a mutation being beneficial.

**Predicted fitness effects of mutation**. Scaled CADD scores for mutations in each evolutionary model were obtained using the online variant annotation tool[25]. Average CADD scores were calculated for mutations in known driver genes and known passenger genes and used to independently validate mutational dynamics in each model. To identify which genes were enriched for function-altering mutations in each evolutionary model, we sampled all mutations with a scaled CADD score of greater than 10, thus retaining the mutations that are most likely to be function-altering, and plotted the intersection of mutations in each gene across the four different evolutionary classes. Finally, to investigate the association between the average pathogenicity of the dominant clone in each evolutionary model, we extracted the clone with the highest variant allele frequency: the dominant clone. The variant allele frequency of the dominant clone was plotted for each CADD interval using bin sizes of 10.

**Distribution of function-altering mutations in genes across evolutionary classes**. An upset plot was used to visualize the distribution of function-altering mutations in genes across the four evolutionary classes. Mutations with a scaled CADD score of greater than 10 were sampled as described above. The upset plot was created using the R package UpSetR.

**Survival analyses**. Survival analyses were performed using the Kaplan–Meier and Cox proportional hazards models. Statistical significance was assessed using a log-rank test and significance was determined at $p < 0.05$.

**Reporting summary**. Further information on research design is available in the Nature Research Reporting Summary linked to this article.

## Data availability

All data included here were collected and published in previous Nature Publications (Abelson et al. Nature 2018). All data were made available to public repositories during that submission. Targeted sequencing data for the discovery cohort are deposited in the European Genome-phenome Archive (http://www.ebi.ac.uk/ega/) under accession number EGAD00001003583. Simulated data has been made available through GitHub at https://github.com/kimskead/popgenArch. Source data for figures are provided with this paper. Source data are provided with this paper.

## Code availability

The reproducible code has been made available through GitHub at https://github.com/kimskead/popgenArch[45].

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

## Acknowledgements

This work was supported by the Ontario Ministry of Research and Innovation to P.A., CIHR Frederick Banting, and Charles Best Canada Graduate Scholarships (Masters and Doctoral) and Terrence Donnelly Center Cecil Yip Award to K.S., and research grants from the Vector Institute for Artificial Intelligence to K.S. and Q.M. P.A. is supported by a Natural Sciences and Engineering Research Council of Canada Award (RGPIN-2019-06813). Q.M. is supported by a Canadian Institute for Advanced Research (CIFAR) Artificial Intelligence Research Chair and core funding from Memorial Sloan Kettering Cancer Center. Genomic data generation was supported by the Canadian Data Integration Center (CDIC) with funds provided by the Government of Ontario and the Government of Canada through Genome Canada and Ontario Genomics (OGI-136).

## Author contributions

P.A. conceived of the study and P.A., S.W., Q.M., and K.S. designed the analyses. L.S., J.D., and S.A. performed experiments and contributed data. A.A.H. and K.S. performed all simulations. K.S., D.S., A.A.H., B.L., V.B., M.A., Q.M., and P.A. performed all analyses. K.S., Q.M., and P.A. wrote the manuscript.

## Competing interests

The authors declare no competing interests.
