## [Peer Review File · Nature Communications]

REVIEWER COMMENTS

Reviewer #1 (Remarks to the Author):

This is a very interesting and provocative paper looking at the evolutionary dynamics underlying CH, a key issue, especially in light of the overselling of CH as a predictor for many things. It is a key intermediate step but not always realized in its worrisome potential.

The premise and the discussion are quite novel and important- namely the dynamics of distinct forces. The authors infer much from only a few time points- they might be right but their data is still on the sparse side with respect to actually measuring the/simulating the dynamics proposed. In this regard, the presentation of the data and its immediate interpretation needs to be done even more cautiously than currently one so. Intuitively, I am with them every step but analytically, the data suggest but do not prove the conclusions. So, please take the care to nuance the presentation of the actual data and simulations.

One thing missing is a clear presentation of the statistical 'power' of the simulations based on the data sources used to generate both the hypotheses and the simulations.

Since this paper straddles between empirical analyses and simulations, more details on the rationale for the decisions made leading to the presented material is critical.

One last philosophical point- if the dynamics of the selective forces are competing and not synchronized, it is interesting that the authors omit the possibility that CH that is eliminated could have an advantage and lead to better outcome (absence of some of serious and dangerous chronic and acute outcomes associated with CH). In this regard, could this be opening the window on preventive biomarkers- or at least temporal biomarkers that could be intermediate and actually- their resolution portend better outcomes?

Reviewer #2 (Remarks to the Author):

The authors describe a deep-learning-based approach to quickly approximate the computationally intensive but well-established Wright-Fisher (WF) population genetic model and discuss the results of the approximation in the context of AML predisposing mutations.

To solve this problem, the authors have designed a multi-task network and trained it on simulated data generated by a WF process. The multi-task heads were optimized to regress the model parameters of the WF processes that generate the simulated data and to classify a set of evolutionary classes defined by the authors.

I like the idea of replacing the computationally intensive WF simulation by a fast approximation, but unfortunately, I don't see enough evidence that results of the WF simulation can be translated into biological findings as discussed in the AML use case, so I cannot recommend the present paper for publication in Nature Comm.

My main criticism is as follows:

- The WF model requires strong assumptions and is defined under the following conditions: A finite but constant population size, random mating, non-overlapping generations and no selection. Except for the finite population, my understanding is that all other assumptions of the WF models are not fulfilled, or how does the WF simulation deal with clonal inference or selection due to environmental factors?
- The choice of model parameters based on the classification into evolutionary classes seems to me

to be quite freely chosen, an ablation study would certainly be appropriate here to investigate the influence of parameter selection on biological findings. This is particularly important with regard to error propagation.

- But I also have my concerns regarding that the model confidence is determined via an ensemble of different model classes, why do the authors not use Bayesian Dropout? At least then the output distributions would be comparable and would not require any adaption by platt-scaling or isotonic regression.
- Another point of criticism is the evaluation of models, it is known that a random CV often leads to an information leak between training and test data. This becomes all the more obvious in the case of the WF simulation, since the same WF model parameters were used to generate the training and the test data set, such a selection bias should be rigorously avoided. This leads to an extreme simplification of tasks and ultimately to overoptimistic results. I therefore do not believe in the generalization capability of the model.

Reviewer #3 (Remarks to the Author):

In general, we think this is a well written paper and an interesting piece of work. We appreciate the fact that the authors put a focus on the effect of the genetic background and the role of non-driver mutations, factors that are often overlooked in models of cancer evolution.

Comments:

0. Please provide **all** of the code for **all** of the simulations and analyses, preferably as a code repository. This should be fully reproducible but as the paper is now it is not reproducible.

The URL you give for SFSCode is just the SFS software. It is not the code that will allow to reproduce your simulations and analyses.

Of course, provide the data you used (not a URL to download the primary data, the actual R data frames you used).

1. **[Major comment?]** Deep Neural Networks (DNNs) are trained using simulated data. Simulations account for several factors that generate variation (coefficients of positive and negative selection, mutation rate, etc.) but leave others out (non-genetic variability, frequency-dependent fitness effects...). If it were the case that these factors were indeed relevant in patient samples, the classification made by DNNs could be unreliable, as data used for training would have been generated using simulations with an incomplete model. This is aggravated by the general matter of DNNs being "black boxes", making it hard to know what underlying factors is classification based upon. How do authors think this potential bias would affect their conclusions? Is there any check that could be made to at least provide some evidence that simulations are similar enough to real evolutionary processes?

2. Along those lines, the authors use Wright-Fisher processes. These are one type of processes, but there is no reason to suppose these are the only ones. Can you comment on this?

Also give reasons for the rest of the parameters of the simulation (e.g., effective population sizes in p. xvii, other parameters in p. xviii and ff)

3. And more along those lines, though this is completely optional: a very different way to do this would be to use ABC (approximate Bayesian computation) with simulations. Can you comment on the differences and advantages of your procedure?

4. In general, some of the explanations in the Results section are poorly discussed or are not derived from the data in a straightforward way. As well, we think that results and discussion often go little beyond merely classifying results with the DNNs.

5. In lines 184 to 187, the authors write that driver mutations are observed in healthy controls under positive selection, and say this would be possible if "the mutations do not alter clonal growth or drive gene function". We think this is weird phrasing: under a traditional "driver vs. passenger" framework, we understand that driver mutations are those that do indeed alter gene function in a way that favors clonal growth. Again, another possible explanation is that the fitness effect of mutations considered drivers is dependent on genetic context due to epistasis (which also applies to their next point of passenger-to-driver mutation ratio being variable between controls and cases).

6. In lines 262-263, the authors suggest that certain clones persisting under a negative selection only regime could be due to a functional threshold at which mutations are tolerated. We are not sure of whether this is really plausible or if tolerated mutations would simply be selected *for* in this case, making it a combined positive and negative selection scenario. Alternative hypotheses for the persistence of these clones could again have to do with epistasis, e.g. a mutation tagged as "driver" could actually be neutral if not in combination with others.

7. At various points through the manuscript (e.g. line 19 in the abstract, line 69 in the introduction...) the authors use the concept "passenger mutation" implying a deleterious fitness effect, which is not the general case as passenger mutations can simply have no effect. We think it would be nice to be more specific (e.g. change "passenger mutations" to "deleterious/neutral passenger mutations" in these instances). We understand that other papers have dealt with the consequences of mildly deleterious "passengers", and similar uses of the word passenger. But we think this contributes to confusion: as far as we know, originally, passenger meant, well, passenger without fitness effect. Later, some authors mentioned that many of those could actually have mildly deleterious consequences, etc, etc. But having terms with single meaning seems important here.

In fact, we were at times confused, and thought that the authors were at times considering compensatory fitness effects between those passengers and the drivers.

So, please, clarify and be consistent in the use of passenger or whatever.

8. In general, we think the text overlooks fact that the fitness increase conferred by driver mutations is conditioned to the genetic context of the cell, not only in terms of non-drivers but also of other drivers. For instance, a clone carrying a driver mutation A and a different clone carrying a driver mutation B could both have high fitness, yet a the combination of A and B could be fitness-reducing or even lethal for a genotype carrying both.

9. Clustering exercise in p. xi and Figure 3b: We find this unnecessary and completely unconvincing. The relevant messages stand on their own without the clustering exercise. Please, remember that clustering is a class discovery tool, not a confirmatory tool. Thus, your use of clustering could, at best, suggest something, but it would not confirm or provide any evidence of anything. We suggest you remove Figure 3b and all references to clustering.

10. p. xviii: In the table, give meaning of parameters: μ , p , s_p , s_n in the first lines of the table itself.

11. At some points in the ms. the grammar is awkward. For example:

- p. ii, lines 22 to 25: "We find that a subset of non-driver genes is enriched for mildly damaging mutations in healthy individuals fitting purifying models of evolution suggesting that mutations in these genes might confer a protective role against disease-predisposing clonal expansions"

- p. vi: "distinct evolutionary model, however, as an overall summary, each model can be collapsed into one of four overarching evolutionary classes as determined by the combination of selection"

- p. xvi: "Indeed, these damaging mutations may hinder the rate that clones drive through the population."

- p. xxxiv: "co-efficient": isn't this "coefficient"?

12. Please always report the output of statistical tests using standard practices: give the statistic, the degrees of freedom, and the p-value; when reporting the p-value, giving figures such as $p < 1e-16$ and other numbers way out in the tails is OK, of course, but $p < 0.0001$ is not.

For example, lots of chi-square are reported without the chi-square statistic or df, some F-tests (e.g., supporting info) are given without the statistic and the numerator and denominator dfs, Wilcoxon tests are reported without the statistic, etc, etc. If any test (among those that can be one or two-tailed) has been performed using the one-tailed version, be explicit and give detailed reasons why it is one-tailed.

13. p. ix shows mu values with 6 significant figures: are these many needed here?

14. Please mention software used for the analyses explicitly in the supporting info too.

15. Using roman numerals to number pages is ... well, unorthodox and cumbersome (for those of us who have trouble with roman numerals to count beyond 40 ---the decimal system has so many fans for a reason :)

Juan Diaz Colunga, Ramon Diaz-Uriarte

REVIEWER COMMENTS

Reviewer #1 (Remarks to the Author):

This is a very interesting and provocative paper looking at the evolutionary dynamics underlying CH, a key issue, especially in light of the overselling of CH as a predictor for many things. It is a key intermediate step but not always realized in its worrisome potential. The premise and the discussion are quite novel and important- namely the dynamics of distinct forces.

We would like to thank the reviewer for pointing out the uniqueness, novelty and importance of our work. We agree that looking at the evolutionary dynamics underlying clonal hematopoiesis is critical in understanding when and how to use clonal hematopoiesis as a predictor for a range of conditions. Further, we fully agree with the reviewer that a better understanding of clonal hematopoiesis is required to better understand and distinguish between actionable and benign clonal expansions.

The authors infer much from only a few time points- they might be right but their data is still on the sparse side with respect to actually measuring the/simulating the dynamics proposed.

We thank the reviewer for their comment here. While our ensemble of deep neural networks were trained and tested entirely on simulated data rather than real data, we recognize that our study was limited to samples from 477 individuals . With that said, each sample is a population and our depth of sequencing coverage is capturing events that are occurring at the cellular level. We are, in principle, looking at 477 distinct populations or evolutionary scenarios. In contrast to other studies, which have largely analyzed datasets that merely capture the mutation with the highest VAF for each individual, we are analyzing the full mutational spectrum for each individual and making inferences of the cellular population.

With respect to the simulations that were used to train and test the model, we performed over five million simulations in total over a broad range of plausible parameters (mutation rate, coefficients of positive and negative selection, and the probability of a mutation being beneficial). Each unique parameter combination is termed an evolutionary model and we generate 2000 instances of each evolutionary model to ensure that we were able to sample a broad range of summary statistics for each evolutionary model. After training our classifier on simulated data, we show that we are able to recover the true evolutionary class for a held out test set of simulated data with high accuracy as demonstrated in our confusion matrix. Subsequently, we applied our classifier to our cohort of pre-leukemic and healthy controls to predict the overarching evolutionary class within each blood population.

In this regard, the presentation of the data and its immediate interpretation needs to be done even more cautiously than currently one so. Intuitively, I am with them every step but analytically, the data suggest but do not prove the conclusions. So, please take the care to nuance the presentation of the actual data and simulations.

We appreciate the reviewers request for us to be cautious in our interpretations and this is related to the reviewers previous point. While we understand that we have made inferences using a single time-point, blood samples were captured retrospectively through linkages to clinical administrative data. So our biological sampling occurred before a known observed “event” and at a range of times prior to the onset of a malignancy. In our discussion, on page 17 we emphasized the need to confirm the proposed conclusions using single cell and longitudinal samples in future studies and we highlight the need for

future work which should explore serial sampling and predictions, but remains outside the scope of this study.

The excerpt from the text is included here: *“Future work with long-read sequencing and longitudinal sampling, is required to experimentally phase somatic mutations genome-wide, will help determine whether such mutations are occurring in the same or different clones thus allowing us to investigate the effects of both individual mutations as well as the epistatic interactions between linked mutations and clonal trajectories.”*

One thing missing is a clear presentation of the statistical 'power' of the simulations based on the data sources used to generate both the hypotheses and the simulations.

Our confusion matrix (Figure 1e) highlights our ability to distinguish between different evolutionary classes. We showed that we were able to distinguish between different evolutionary classes with an accuracy of 86.1%. We specifically chose an approach that would allow us to evaluate the certainty associated with each evolutionary class prediction in our classifier. As mentioned in a previous answer, we performed over five million simulations to train our model to recognize a broad range of evolutionary settings. . After training, our classifier recovers the true evolutionary class for a held-out from a similar set of evolutionary models with very high accuracy (Figure 1e). As such, we are very well-powered.

Since this paper straddles between empirical analyses and simulations, more details on the rationale for the decisions made leading to the presented material is critical.

We would like to thank the reviewer for these comments. We have updated our methods to more explicitly define the rationale for the parameter selection in the *Methods* section of the paper. The range of model parameters were chosen to correspond to biologically realistic values (such as mutation rates and population sizes) but also were selected to be as wide-ranging as possible so that we do not bias our conclusions by simulating too small a parameter space. We performed millions of simulations to explore as wide a range of selection coefficients (positive, negative and neutral) in order to capture the most likely model fitting the data. To demonstrate that our parameters did indeed generate biologically realistic values, and values which correspond to our empirical data, we have added a figure (*Supplementary Figure 1*) to show overlap between the distribution of simulated summary statistics and real data.

We have included the Figure from Page 37 of the manuscript below.

Supplementary Figure 1. Distribution of summary statistics from simulated data and observed data. t-Distributed Stochastic Neighbor Embedding (t-SNE) mapping the distribution of 16 summary statistics for simulated and observed blood populations. Points were sampled from four different evolutionary classes (total sample size: $N = 20,000$) and are shown in colour (blue, green, orange and red). Observed summary statistics from the EPIC cohort are shown in black). Importantly, points representing summary statistics for patient data from the EPIC cohort are contained within the spread of the distribution of the simulated statistics indicating that there are within the distribution of summary statistics produced by the simulations.

One last philosophical point- if the dynamics of the selective forces are competing and not synchronized, it is interesting that the authors omit the possibility that CH that is eliminated could have an advantage and lead to better outcome (absence of some of serious and dangerous chronic and acute outcomes associated with CH). in this regard, could this be opening the window on preventive biomarkers- or at least temporal biomarkers that could be intermediate and actually- their resolution portend better outcomes?

We would like to thank the reviewer for this insightful comment. We completely agree that CH that is eliminated could have an advantage and lead to a better outcome. To illustrate this, we have included Kaplan-Meier curves showing AML-free survival, defined as the number of days between blood sample collection and AML diagnosis, last follow up or death. Our survival curves have been stratified by overarching evolutionary class.

We have included the following text and figure on pages 14/15 and 35 of the manuscript, respectively.

Results section (page 14 and 15): *Indeed, we find that ARCH occurring in the absence of positive selection, that is individuals who fit negative or neutral classes of evolution, is associated with a lower risk of progression to AML compared to individuals who have signatures of positive selection (log rank test, $p = 2e-04$)(Figure 4a). We find that individuals who fit combination classes of evolution have an approximately two-fold increased risk of progressing to AML compared to individuals fitting neutral models of evolution (hazard ratio, 2.43; 95% confidence interval, 1.45–4.06, Figure 4b). Owing to the small number of individuals fitting positive only classes of evolution, we cannot infer if the negative selection acting on passenger mutations in individuals fitting combination classes of evolution reduces risk of progressing to AML. However, a scenario where multiple clones compete for dominance, thus maintaining clones at*

intermediate frequencies, would explain the greater risk conferred to individuals fitting combination classes of evolution.

Addition to figures (page 35):

Figure 4

Figure 4. a) Kaplan–Meier curves of AML-free survival among EPIC participants. Survival is defined as the time between blood sample collection and diagnosis (for cases), for control survival is right-censored at last follow-up. Survival curves are stratified according to evolutionary class. **b)** Forest Plot of risk of AML development across evolutionary classes. Each row indicates hazard ratios associated with each evolutionary class (reference = neutral class). The horizontal lines indicate 95% confidence interval for each class.

Additionally, to address this point, we have edited/ included text on page 17, to state that: “*The presence of negative selection acting on passenger mutations in the presence of driver mutations offers a potential explanation for why some individuals who harbour driver mutations do not progress to disease. However, negative selection appears to play an important role in lowering the overall pathogenicity of the mature blood cell pool, thus maximizing the fitness of the individual; a finding which suggests that there could be an advantage to retaining mildly damaging mutations in cellular populations if they confer a protective effect in the presence of a driver mutation. Indeed, our survival analyses demonstrate that the evolutionary forces shaping ARCH are the same in that patterns of clonal hematopoiesis fitting negative or neutral classes of evolution are typically associated with lower risk of progression to AML.*”

Reviewer #2 (Remarks to the Author):

The authors describe a deep-learning-based approach to quickly approximate the computationally intensive but well-established Wright-Fisher (WF) population genetic model and discuss the results of the approximation in the context of AML predisposing mutations. To solve this problem, the authors have designed a multi-task network and trained it on simulated data generated by a WF process. The multi-task heads were optimized to regress the model parameters of the WF processes that generate the simulated data and to classify a set of

evolutionary classes defined by the authors. I like the idea of replacing the computationally intensive WF simulation by a fast approximation,

We would like to thank the reviewer for their positive comments on our approach. We would like to clarify that we are not replacing simulations with an approximation of the Wright Fisher (WF) process. We are rather replacing the inference process for a family of forward simulations of clonal (somatic) evolution subject to a number of deterministic and stochastic evolutionary forces. We have clarified the language in the description to make this more clear. The simulations are a critical step in training our network and in validating its accuracy on how well we are able to recover different evolutionary scenarios. Typically inference would be done with Approximate Bayesian Computation (ABC), which is also approximate inference, here we are using a recognition model.

but unfortunately, I don't see enough evidence that results of the WF simulation can be translated into biological findings as discussed in the AML uses case, so I cannot recommend the present paper for publication in Nature Comm.

As indicated in our response listed above as we believe that there is some confusion about our approach. As mentioned we do not create an approximation of the WF, nor do we attempt to fit to a purely WF model. We would like to thank the reviewer for highlighting this as an area that required further clarification. In order to avoid confusion, we have clarified our methods and introduction to better illustrate our approach and have removed reference to the WF process throughout the manuscript.

My main criticism is as follows:

- *The WS model requires strong assumptions and is defined under the following conditions: A finite but constant population size, random mating, non-overlapping generations and no selection. Except for the finite population, my understanding is that all other assumptions of the WF models are not fulfilled, or how does the WF simulation deals with clonal inference or selection due environmental factors?*

We thank the reviewer for their comments here and we have updated the text to be more clear. While SFS-code as a population genetic simulator uses the WF process as a starting point for forward simulations, including the assumptions as they are mentioned above, it is not limited to those assumptions. These forward simulation approaches are used universally in evolutionary biology, in both clonal and non-random mating systems (e.g. Hough et al. Genetics 2017, Buffalo & Coop. PNAS 2020). As the reviewer points out, we are also simulating non-neutral processes, including varying selection coefficients (as well as allowing for neutral mutations).

We also note that the reviewer seems to be confused about our simulation process: we are simulating a diploid asexually reproducing system, so there is no mating (random or otherwise), nor is there recombination.

We have updated our text to clarify that we are doing forward simulations that simulate clonal processes including mutations that are acted upon by selection (non-WF processes). The reviewer is correct that we have ignored population size changes such as population growth; we anticipate that if a mutation is beneficial, the clone harboring that mutation will grow in frequency within the tissue at a rate dependent on the strength of selection acting on that mutation. To address this concern, we make these statements more obvious in the main text with a citation (Lee-Six, H., et al. 2018; the Campbell group showed that HSC population size remains steady at human adolescence). With respect to environmental factors, we

recognize that environmental factors can often come into play, however, that is outside the scope of this study and we do not have the data to justify parameterizing environmental factors presently.

- *The choice of model parameters based on the classification into evolutionary classes seems to me to be quite freely chosen, an ablation study would certainly be appropriate here to investigate the influence of parameter selection on biological findings. This is particularly important with regard to error propagation.*

Please refer to our response to reviewer #1. The range of model parameters were chosen intentionally to correspond to biologically realistic values. We wanted to explore as wide a range of selection coefficients (positive, negative and neutral) as well as mutation rates in order to capture the most likely model fitting the data. As described above, we have expanded on our *Methods* section to better illustrate why the specific parameter ranges were selected. Also please see Supplementary Figure 1 in our response above showing that the patient data lies within the data distributions generated by our simulations.

- *But I also have my concerns regarding that the model confidence is determined via an ensemble of different model classes, why do the authors not use Bayesian Dropout? At least then the output distributions would be comparable and would not require any adaption by platt-scaling or isotonic regression.*

This concern may also arise out of confusion here regarding our approach. Our model is not an ensemble of different model classes, rather, as explained in text from the original manuscript, we are using an ensemble of classifiers all of which have identical architectures but that are trained on different subsets of the data. We evaluated the reliability of our classifier and it is well calibrated as indicated in Response to Reviewers Figure 1 (below).

Response to Reviewers Figure 1. Calibration figure for classifier. Line (red) compares output probability assigned to most probable evolutionary class by our network to the fraction of held-out cases where the predict class matched the actual one. Dotted black line indicates perfect calibration.

Recent work (e.g. Lakshminarayanan et al. 2017 and works cited within) shows that ensemble-based approaches are able to perform better than variational Bayesian approaches (like Bayesian dropout) in estimating uncertainty and in model calibration because they capture multiple modes of the function space (see, e.g., <https://arxiv.org/abs/1912.02757>) as opposed to the single mode captured by dropout

- *Another point of criticism is the evaluation of models, it is known that a random CV often leads to an information leak between training and test data. This becomes all the more obvious in the case of the WF simulation, since the same WF model parameters were used to generate the training and the test data set, such a selection bias should be rigorously avoided. This leads to an extreme simplification of tasks and ultimately to overoptimistic results. I therefore do not believe in the generalization capability of the model.*

We agree that data hygiene and information leakage is a problem in many ML applications. It is not a problem here. First, there is no selection bias, this is a misunderstanding of the learning problem: we are trying to predict the parameter settings used to generate the data, in other words, these are target values (or outputs) of the network not inputs to it. The test set is generated completely independently of the training set -- and there are 2000 independent sets of data points (divided among the training and test set) per parameter setting.

Reviewer #3 (Remarks to the Author):

In general, we think this is a well written paper and an interesting piece of work. We appreciate the fact that the authors put a focus on the effect of the genetic background and the role of non-driver mutations, factors that are often overlooked in models of cancer evolution.

We would like to thank the reviewers for their positive comments and for the detail with which the paper was reviewed. The comments were very helpful.

Comments:

*0. Please provide ***all*** of the code for ***all*** of the simulations and analyses, preferably as a code repository. This should be fully reproducible but as the paper is now it is not reproducible.*

The URL you give for SFScode is just the SFS software. It is not the code that will allow to reproduce your simulations and analyses.

Of course, provide the data you used (not a URL to download the primary data, the actual R data frames you used).

We have made our code and data available through a link in the paper (page 20).

*1. ***[Major comment?]*** Deep Neural Networks (DNNs) are trained using simulated data. Simulations account for several factors that generate variation (coefficients of positive and negative selection, mutation rate, etc.) but leave others out (non-genetic variability, frequency-dependent fitness effects...). If it were the case that these factors were indeed relevant in patient samples, the classification made by DNNs could be unreliable, as data used for training would have been generated using simulations with an incomplete model. This is aggravated by the general matter of DNNs being "black boxes", making it hard to know what underlying factors is classification based upon. How do authors think this potential bias would affect their conclusions? Is there any check that could be made to at least provide some evidence that simulations are similar enough to real evolutionary processes?*

We thank the reviewer for this comment.

The nature of model fitting and parameter testing is that one cannot model (or simulate) all possible evolutionary scenarios. However, in this case, we attempted to evaluate a very large set of population scenarios and captured features that would be most relevant in this biological context. In the future, we could naturally simulate an even broader range of evolutionary scenarios, including frequency dependence etc., and these models may indeed be relevant but in cancer. There has thus far been little evidence that these models are relevant in early cancer development, however, in the context of post-treatment, a rare clone can potentially escape treatment and its rareness may provide a fitness advantage – however we are not looking at post-treatment data, our dataset is entirely derived from pre-leukemic individuals and controls.

And again, with respect to environmental factors, these are indeed interesting (see response Reviewer 2). There may be situations where we cannot fit data to a model because the model is not complicated enough

to incorporate all possible scenarios. Often, environmental factors can come into play and we recognize that, however we do not have data that can justify identifying specific parameters to model/simulate.

To evaluate if our simulations are similar to real evolutionary scenarios, we have compared the distribution of summary statistics from our simulated evolutionary scenarios to the summary statistics from our real data (please refer to response to Reviewer 1).

The reviewer's concern is actually two separate questions: (1) are the forward simulations sufficiently realistic to be useful?, and (2) does our DNN perform sufficiently accurate inference to recover the parameters of a given simulation? In regards to the first question, we have selected parameter ranges that are both realistic and justifiable. The manuscript has now been updated to clarify these points. Further support is provided by the fact that the summary statistics for real patients lie within high probability regions of the distributions of summary statistics for simulated patients (see Supplementary Fig 1 above). Furthermore, our model replicates the expected trends in patient data (e.g., pre-leukemics have more positive selection than controls). In regards to the second question, a DNN model for inference is no more a black box than ABC inference procedures typically used in this field. Nonetheless, here it is possible to directly assess inference performance (as our confusion matrix does). Coupled with Supplementary Figure 1, we can confidently say that our model is performing accurate inference of evolutionary class for in-distribution samples (with the exception of 20% of the negative and neutral samples, a well-known difficult distinction).

2. Along those lines, the authors use Wright-Fisher processes. These are one type of processes, but there is no reason to suppose these are the only ones. Can you comment on this?

As raised by the second reviewer, and addressed in our answer, our “Wright-Fisher” model is only WF in the sense that we are performing forward-in-time simulations, rather than say a coalescent approach (going backwards in time). We simulate clonal non-neutral processes which in themselves are not WF. The simulation approach incorporated is flexible enough that we can consider a wide-range of evolutionary scenarios. As such, we have removed reference to Wright-Fisher processes and instead refer to forward simulations.

Also give reasons for the rest of the parameters of the simulation (e.g., effective population sizes in p. xvii, other parameters in p. xviii and ff)

Please refer to our response to Reviewer One. We have elaborated on this in our Methods.

3. And more along those lines, though this is completely optional: a very different way to do this would be to use ABC (approximate Bayesian computation) with simulations. Can you comment on the differences and advantages of your procedure?

We agree with the reviewers that this would be an alternative option through which to approximate the evolutionary parameters in blood. Initially, we explored an Approximate Bayesian Approach and were unable to obtain the same accuracy in discriminating amongst evolutionary classes given the summary statistics used. One advantage of an ABC approach is that you are able to obtain a posterior distribution of plausible scenarios rather than one best fit option which allows one to better understand the certainty of your prediction. Accordingly, we employed an ensemble-based approach in order to obtain estimates

of the certainty in our predictions by looking at the distributions of estimates of the parameters across the ensemble.

4. In general, some of the explanations in the Results section are poorly discussed or are not derived from the data in a straightforward way. As well, we think that results and discussion often go little beyond merely classifying results with the DNNs.

We would like to thank the reviewers for raising this point. Based on this, and comments from reviewer one, we have emphasized the need to confirm the proposed conclusions using single cell and longitudinal samples in future studies in our discussion section (see response to Reviewer One). However, the major contribution of this paper, that the effect of passenger mutations cannot be ignored, is demonstrated through our classification-based approach.

5. In lines 184 to 187, the authors write that driver mutations are observed in healthy controls under positive selection, and say this would be possible if "the mutations do not alter clonal growth or drive gene function". We think this is weird phrasing: under a traditional "driver vs. passenger" framework, we understand that driver mutations are those that do indeed alter gene function in a way that favors clonal growth. Again, another possible explanation is that the fitness effect of mutations considered drivers is dependent on genetic context due to epistasis (which also applies to their next point of passenger-to-driver mutation ratio being variable between controls and cases).

We agree with the reviewers that this phrasing is confusing and we have updated the text accordingly. Page 9: "It is possible that these individuals do not progress to disease if driver mutations are arising in competing clones, thus preventing one clone from rising to dominance."

*6. In lines 262-263, the authors suggest that certain clones persisting under a negative selection only regime could be due to a functional threshold at which mutations are tolerated. We are not sure of whether this is really plausible or if tolerated mutations would simply be selected *for* in this case, making it a combined positive and negative selection scenario. Alternative hypotheses for the persistence of these clones could again have to do with epistasis, e.g. a mutation tagged as "driver" could actually be neutral if not in combination with others.*

The reviewers point is fair and is the rationale for why we have simulated scenarios where both positive and negative mutations are allowed to accrue (combination). We agree with the reviewers that, In the combination class, it may be that negative mutations are linked to positive mutations and the overall net effect is that they are not removed. However, we expect that strongly deleterious mutations would be quickly removed from the population in the negative only class and that weaker selection acting on mildly damaging mutations might act less efficiently resulting in those mutations being observed in the population. As such, we have updated the relevant sentence on page 13 to read: "Clones which persist in the negative-only model could indicate a functional threshold at which mutations are not efficiently removed by selection and continue to segregate in the population."

7. At various points through the manuscript (e.g. line 19 in the abstract, line 69 in the introduction...) the authors use the concept "passenger mutation" implying a deleterious fitness effect, which is not the general case as passenger mutations can simply have no effect. We think it would be nice to be more specific (e.g. change "passenger mutations" to "deleterious/neutral passenger mutations" in these instances). We understand that other papers have dealt with the consequences of mildly deleterious "passengers", and similar uses of the

word passenger. But we think this contributes to confusion: as far as we know, originally, passenger meant, well, passenger without fitness effect. Later, some authors mentioned that many of those could actually have mildly deleterious consequences, etc, etc. But having terms with single meaning seems important here.

In fact, we were at times confused, and thought that the authors were at times considering compensatory fitness effects between those passengers and the drivers.

So, please, clarify and be consistent in the use of passenger or whatever.

We agree with the authors that 1) passenger mutations originally meant mutations without fitness effect and are commonly used as such and that 2) other papers have referred to mildly deleterious mutations as passenger mutations. In this paper, we consider passenger mutations to be any mutation that does not drive clonal growth (ie not a driver mutation) and which might be swept to a higher frequency as a result of selection acting on a driver mutation. We have clarified our definition on page 4 of the manuscript to state that “For the remainder of this paper, we will refer to mutations accumulating in non-driver genes as “passenger mutations”. Many passenger mutations are neutral and will have no impact on the fitness of the clone, however, some passenger mutations will be mildly damaging, and should they occur within a clone carrying a driver mutation, could impact the rate of expansion of the driver-harboring clones (Figure 1a) (15).” Where possible, we have clarified whether we are referring to mildly damaging or neutral passenger mutations.

8. In general, we think the text overlooks fact that the fitness increase conferred by driver mutations is conditioned to the genetic context of the cell, not only in terms of non-drivers but also of other drivers. For instance, a clone carrying a driver mutation A and a different clone carrying a driver mutation B could both have high fitness, yet a the combination of A and B could be fitness-reducing or even lethal for a genotype carrying both.

The reviewer makes a good point. We are assuming additive fitness effects. We agree that there could be epistatic effects but this would need a lot more data to detect, and our model is already more complete than any other model of CH. We now acknowledge this limitation in the discussion section. Nonetheless, our additive model is detecting interesting effects. On page 17, we state that: “Future work with long-read sequencing and longitudinal sampling, is required to experimentally phase somatic mutations genome-wide, will help determine whether such mutations are occurring in the same or different clones thus allowing us to investigate the effects of both individual mutations as well as the epistatic interactions between linked mutations and clonal trajectories.”

9. Clustering exercise in p. xi and Figure 3b: We find this unnecessary and completely unconvincing. The relevant messages stand on their own without the clustering exercise. Please, remember that clustering is a class discovery tool, not a confirmatory tool. Thus, your use of clustering could, at best, suggest something, but it would not confirm or provide any evidence of anything. We suggest you remove Figure 3b and all references to clustering.

We agree with the reviewer that the clustering, at best, suggests that certain genes might be subject to various selective pressures based on the frequency at which function altering mutations are distributed across genes in different evolutionary classes. Based on the reviewers recommendation, we have removed Figure 3b and have replaced it with the following UpSet plot to showing the distribution of mutations across genes in the different evolutionary classes:

Figure 3b. Distribution of functional mutations in genes across evolutionary classes. UpSet plot shows the distribution of function-altering mutations (CADD > 10) falling in genes across patients in different evolutionary classes. The total number of genes mutated in each evolutionary class is shown on the left. The dark circles indicated classes with overlapping genes and the connecting bar indicated multiple overlapping genes.

10. p. xviii: In the table, give meaning of parameters: μ , p , s_p , s_n in the first lines of the table itself.

We have added in the names of each parameter in the first line of the table. The updated table can be found on page 20 of the manuscript and is included below.

Class	Mutation rate (μ)	Probability of a mutation being a driver mutation (p)	Coefficient of positive selection (s_p)	Coefficient of negative selection (s_n)
Neutral	All	0	0	0
Positive Selection	All	$0 < p < 0.5$; $p=1$	$0 < s_p < 0.05$	0
Negative Selection	All	0	0	$0 < s_n < 0.05$
Combination	All	$0 < p < 1$	$0 < s_p < 0.05$	$0 < s_n < 0.05$

11. At some points in the ms. the grammar is awkward. For example:

We would like to thank the reviewer for highlighting these sections of the text for revision. We have included our revisions below.

- p. ii, lines 22 to 25: *"We find that a subset of non-driver genes is enriched for mildly damaging mutations in healthy individuals fitting purifying models of evolution suggesting that mutations in these genes might confer a protective role against disease-predisposing clonal expansions"*

Page 2: *"We find that a subset of genes are enriched for mutations in individuals with signatures of purifying selection suggesting that mutations in these genes might confer a protective role against disease-predisposing clonal expansions."*

- p. vi: *"distinct evolutionary model, however, as an overall summary, each model can be collapsed into one of four overarching evolutionary classes as determined by the combination of selection"*

Page 6: *"Each unique combination of four parameters corresponds to a distinct evolutionary model. However, each model can be collapsed into one of four overarching evolutionary classes: neutral (no selection), positive selection only, negative selection only, and combination models which allow for the accumulation of mutations subject to both positive and negative selection within the same cellular population (see Online Methods)."*

- p. xvi: *"Indeed, these damaging mutations may hinder the rate that clones drive through the population."*

Page 16: *"Negative selection acting on linked passenger mutations might slow the rate of clonal expansions."*

- p. xxxiv: *"co-efficient": isn't this "coefficient"?*

The term *"co-efficient"* has been replaced with the term *"coefficient"* throughout the paper.

12. Please always report the output of statistical tests using standard practices: give the statistic, the degrees of freedom, and the p-value; when reporting the p-value, giving figures such as $p < 1e-16$ and other numbers way out in the tails is OK, of course, but $p < 0.0001$ is not.

We have made the necessary corrections throughout the document.

For example, lots of chi-square are reported without the chi-square statistic or df, some F-tests (e.g., supporting info) are given without the statistic and the numerator and denominator dfs, Wilcoxon tests are reported without the statistic, etc, etc. If any test (among those that can be one or two-tailed) has been performed using the one-tailed version, be explicit and give detailed reasons why it is one-tailed.

We have updated the test on page 20 to state that: *"All Wilcoxon rank sum tests are two-sided unless otherwise specified."*

13. p. ix shows mu values with 6 significant figures: are these many needed here?

We agree that six significant figures are not necessary and have edited the text accordingly. The relevant text on Page 9 now reads: “...(mean mutation rate: $\mu = 1.2e-10$ per bp per division) compared to healthy controls (mean mutation rate: $\mu = 1.1e-10$ per bp per division)...”

14. Please mention software used for the analyses explicitly in the supporting info too.

We have listed the software used for the analyses explicitly in the *Methods* section. Page 20: *Software for statistical analyses. Following the simulation of data using SFSCode, all training and testing of deep neural networks, analyses and figures were performed in the R statistical programming environment. Libraries that are required include plyr (30), dplyr (31), UpSetR (32), Rtsne (33), lsmeans (34), forestmodel (35), survival (36), survminer (37) and keras (38). Figures were generated using ggplot2 (39), ggpubr (40), ggplotify (41) and patchwork (42). All Wilcoxon rank sum tests are two-sided unless otherwise specified. The reproducible code has been made available in Supplementary Code.*

15. Using roman numerals to number pages is ... well, unorthodox and cumbersome (for those of us who have trouble with roman numerals to count beyond 40 ---the decimal system has so many fans for a reason :)

We have changed the page numbering format.

REVIEWER COMMENTS

Reviewer #1 (Remarks to the Author):

The authors have thoroughly addressed my queries and most of the other reviewers, though the issue of addressing mixing is not really relevant.

The new language addressing the nuances is improved

Reviewer #2 (Remarks to the Author):

I thank the authors for their explanations and have to admit that I really misunderstood the simulation process. With the explanations in the text the intention is better described and most of my criticism points are sufficiently addressed.

However, I am still not convinced that the model evaluation is so correct.

The authors simulated hematopoietic populations across a grid of parameters: mutation rate (μ), the probability of a mutation being beneficial (p), coefficient of positive selection, and coefficient of negative selection. Of the data generated in this way, 90% and 10% were randomly chosen for training and testing, respectively. Therefore, there is a clear information leak between training and test data, which leads to overoptimistic results regarding the predicted simulation parameters, since a simulated data set generated with the same parameters to be predicted already existed during training.

To test the model performance under real conditions, a leave parameter range-out cross-validation has to be performed. For example, the range of each simulation parameter can be divided into 4 bins (each bin corresponds to one fold per parameter), which leads to 16 folds in total. For the training process only 15 folds should be used to predict the 16th fold.

Reviewer #3 (Remarks to the Author):

We haven't had the opportunity to look at this with the depth we would have liked, nor will we have time to look at it carefully within the next weeks, but the review is long overdue, so we thought it better to provide these comments even if, as we say, we haven't had a chance to look at this with the depth it deserves.

The authors provide new material showing that the summary statistics from observed data fall within the spread of the summary statistics of the simulated data for their choice of parameter values. While this is indeed compatible with the underlying evolutionary dynamics of the simulations resembling real processes (or at least capturing their key determinants), it does not constitute proof on itself that their model is accurate enough as to justify the use of simulation-trained DNNs on biological observations. Many models could produce summary statistics within the simulated values while still being far from realistic.

Their confusion matrix provides an upper bound on inference performance when dealing with observed data, but it remains unclear what this performance is in practice.

Regarding whether the evolutionary models are realistic enough to allow for classifying the biological data, it seems again that the authors have "bypassed" the question itself by calling "clonal haploid populations evolving forward in time" what they used to call (incorrectly) "Wright-Fisher processes". But the underlying issues are still there?

Additionally, the authors acknowledge the limitation coming from the fact that their model assumes additive fitness effects. While this limitation remains and is, in our opinion, a major drawback, we do think that it is now more clearly acknowledged and discussed in the manuscript. Similarly, while some of our concerns regarding the extent to which the models can be used to draw conclusions on biological observations remain, the text has been improved to a point where these concerns have been more clearly and explicitly addressed and discussed.

In general, the instances where the authors have rephrased/elaborated on their statements (particularly regarding the use of the term "passenger mutations") have helped improve the clarity of the text and make it more consistent. Furthermore, our concerns regarding statistical rigor and code/materials availability have been fully addressed.

To conclude: Our biggest general concern was that the authors use simulated data to train neural network to make inferences about biological data without guarantees that the simulated processes are close enough to the real data themselves. Has this concern been satisfactorily addressed? We are hesitant to say so. We think the authors, in their responses, have "navigated around" the issue without addressing it directly. On the other hand, there might not be much else that can be directly done to fully address this issue. And we think that the changes in the text really improve the ms. and make all of these limitations explicit.

REVIEWER COMMENTS

Reviewer #1 (Remarks to the Author):

The authors have thoroughly addressed my queries and most of the other reviewers, though the issue of addressing mixing is not really relevant.

The new language addressing the nuances is improved

We would like to thank the reviewer for their helpful comments throughout the review process and are glad to hear that the new language addressed their concerns.

Reviewer #2 (Remarks to the Author):

I thank the authors for their explanations and have to admit that I really misunderstood the simulation process. With the explanations in the text the intention is better described and most of my criticism points are sufficiently addressed.

Thank-you for letting us know that the new text addressed your concerns and clarified our intention.

However, I am still not convinced that the model evaluation is so correct.

The authors simulated hematopoietic populations across a grid of parameters: mutation rate (μ), the probability of a mutation being beneficial (p), coefficient of positive selection, and coefficient of negative selection. Of the data generated in this way, 90% and 10% were randomly chosen for training and testing, respectively. Therefore, there is a clear information leak between training and test data, which leads to overoptimistic results regarding the predicted simulation parameters, since a simulated data set generated with the same parameters to be predicted already existed during training.

To test the model performance under real conditions, a leave parameter range-out cross-validation has to be performed. For example, the range of each simulation parameter can be divided into 4 bins (each bin corresponds to one fold per parameter), which leads to 16 folds in total. For the training process only 15 folds should be used to predict the 16th fold.

The reviewer is concerned that our deep neural network ensemble (DNNE) has overfit the training set and memorized some of the parameter settings. We agree that this is an important general concern, and we thank the reviewer for raising it. To address it, we checked for overfitting in a more efficient and more direct way than the reviewer proposed; and we find no evidence of it or “information leakage” impacting our results. If anything, the model performs slightly worse at classifying test set examples that are similar to ones in the training set. Our explanation for why our direct approach is appropriate, and how we tested for overfitting is detailed below.

ROLE OF THE DNNE: First, to clarify, the focus of this manuscript is not the DNNE, we are simply using it for amortized inference as a replacement for ABC. This approach, amortized inference replacing ABC, is by now a well-established technique. As such, we hesitate to devote much attention to it in our manuscript.

ISSUE OF INFORMATION LEAKAGE: The key question raised by the reviewer is whether our DNNE generalizes to parameter settings, Y , that don't appear in our training set. Note

that because we are doing amortized inference, we normally would not be worried about a substantial overlap between the values of Y present in training and test set, indeed this is something that we would hope to see. Also, because we are sampling data from the evolutionary simulation, we can generate a lot of training data to train the model -- and indeed, we have thousands of samples of inputs X for each setting of Y. So, we would normally also not expect overfitting. What the reviewer raises as a point of concern is that we are sampling Y from a grid rather than from a more uniform distribution on Y.

Rather than re-training the model, as proposed by the reviewer, we opted to test for overfitting directly: by generating new evolutionary data from parameter settings not used in the training set. This approach matches the reviewer’s suggestion in spirit, while also explicitly testing the model that we applied to real data. Under the reviewer’s proposed cross-validation scheme, we would never actually be evaluating the generalization ability of the ultimate model used on the real data.

First, however, as a preliminary test for overfitting and to explicitly remove clear, potential cases of “information leakage”, we removed from the test set any data points which exactly matched a data point in the training set. Surprisingly, when doing so, we actually observe a small increase in test set classification accuracy – suggesting those data points identical between the training and test set are actually more likely to be ambiguous with regards to evolutionary class. We have modified Figure 1e to show the performance of our ensemble when these matching data points are removed from the test set.

True Class	Neutral	0% n=0	0% n=3	16.6% n=1740	83.4% n=8728
	Negative	0% n=0	0.2% n=7	80.6% n=3415	19.3% n=817
	Combination	0% n=0	97.4% n=46138	2.1% n=983	0.5% n=233
	Positive	99.7% n=11864	0.2% n=24	0.1% n=12	0% n=0
		Positive	Combination	Negative	Neutral
		Predicted Class			

Figure 1e. Classification performance for simulated evolutionary classes. We obtain a high classification accuracy across evolutionary classes (94.8%). Positive and combination classes are predicted with 99.7% and 97.4%, respectively. We observe a reduction in accuracy in neutral (80.6%) and negative (83.4%) classes of evolution.

Additionally, we included the following text in our methods (page 22):

(Methods, page 22). Any instance where there are identical sets of summary statistics in the training and test set were removed from the test set prior to evaluation in order to eliminate information leakage.

To explicitly test generalization of the model to parameter settings not seen in the training set, we generated a new set of simulated data where we shifted the parameter grid points

away from those used in training. We then evaluated the ability of the model to predict evolutionary classes on these new parameter combinations. We achieved a highly similar degree of accuracy on simulations using these new parameter settings.

We have included Supplementary Figure 3 to show the performance of our ensemble in identifying true evolutionary classes in a model which it has not been previously exposed to.

True Class	Neutral	0% n=0	0.2% n=6	16.4% n=592	83.3% n=2999
	Negative	0% n=0	1.1% n=45	80.6% n=3222	18.1% n=722
	Combination	0% n=0	99.6% n=18790	0.4% n=71	0% n=0
	Positive	97.3% n=3681	2.6% n=100	0.1% n=4	0% n=0
		Positive	Combination	Negative	Neutral
		Predicted Class			

Supplementary Figure 3. Classification performance on novel parameter combinations. We tested our model’s performance on a novel set of simulated data generated from novel parameter combinations. We find that we are able to achieve similar degrees of accuracy in evolutionary class predictions with positive and combination classes are predicted with 97.3% and 99.6%, respectively. Similarly, we observe a similar minor reduction in accuracy in discriminating between neutral and negative evolutionary classes with our classifier achieving accuracies of (80.6%) and (83.3%), respectively.

Additionally, we have included the following text:

(Results, page 7). To ensure that our model was able to perform well on data from evolutionary parameters not included in our training data, we generated a novel set of simulations using new parameter combinations that do not appear in the training set. We find that we are able to achieve a similar degree of accuracy in evolutionary class prediction (Supplementary Figure 3).

(Methods, page 22) To ensure that our classifier was able to generalize well to parameter combinations not included in the training data, we generated an additional test set of simulated data wherein the gamma distributions from which parameters are drawn were modified thus generating novel values and parameter combinations not included in the training and test set.

Reviewer #3 (Remarks to the Author):

We haven't had the opportunity to look at this with the depth we would have liked, nor will we have time to look at it carefully within the next weeks, but the review is long overdue, so we thought it

better to provide these comments even if, as we say, we haven't had a chance to look at this with the depth it deserves.

The authors provide new material showing that the summary statistics from observed data fall within the spread of the summary statistics of the simulated data for their choice of parameter values. While this is indeed compatible with the underlying evolutionary dynamics of the simulations resembling real processes (or at least capturing their key determinants), it does not constitute proof on itself that their model is accurate enough as to justify the use of simulation-trained DNNs on biological observations. Many models could produce summary statistics within the simulated values while still being far from realistic.

Their confusion matrix provides an upper bound on inference performance when dealing with observed data, but it remains unclear what this performance is in practice.

Regarding whether the evolutionary models are realistic enough to allow for classifying the biological data, it seems again that the authors have "bypassed" the question itself by calling "clonal haploid populations evolving forward in time" what they used to call (incorrectly) "Wright-Fisher processes". But the underlying issues are still there?

Additionally, the authors acknowledge the limitation coming from the fact that their model assumes additive fitness effects. While this limitation remains and is, in our opinion, a major drawback, we do think that it is now more clearly acknowledged and discussed in the manuscript. Similarly, while some of our concerns regarding the extent to which the models can be used to draw conclusions on biological observations remain, the text has been improved to a point where these concerns have been more clearly and explicitly addressed and discussed.

In general, the instances where the authors have rephrased/elaborated on their statements (particularly regarding the use of the term "passenger mutations") have helped improve the clarity of the text and make it more consistent. Furthermore, our concerns regarding statistical rigor and code/materials availability have been fully addressed.

To conclude: Our biggest general concern was that the authors use simulated data to train neural network to make inferences about biological data without guarantees that the simulated processes are close enough to the real data themselves. Has this concern been satisfactorily addressed? We are hesitant to say so. We think the authors, in their responses, have "navigated around" the issue without addressing it directly. On the other hand, there might not be much else that can be directly done to fully address this issue. And we think that the changes in the text really improve the ms. and make all of these limitations explicit.

We would like to thank the reviewers for their re-review especially considering their time pressures and, again, for their initial, helpful comments that helped us to improve the paper. We are happy to hear that the changes we made improved the manuscript.

The reviewers highlight a remaining concern regarding if our evolutionary model reflects reality. Our approach of using simulated populations evolving through a range of processes is a well-established tool and has led to many advances in our understanding of clonal, and cancer, evolution (see references 3,10,11). Many of these studies have used evolutionary scenarios that are more limited than what we have evaluated (such as purely neutral or positive evolutionary scenarios). Our work highlights the role that negative selection, which is frequently overlooked, might play in shaping clonal dynamics leading up to overt malignancy and, as such, already expands the scope of plausible evolutionary

scenarios governing clonal dynamics beyond what has already been studied in clonal hematopoiesis.

Specifically, the reviewers ask if we can extend our modelling approaches to incorporate non-additive events, however, they also note that this might not be possible. We agree with the reviewers that, while this is interesting, it is impossible for us to address given the size of the dataset and the available phenotypic data. The dataset we used which, despite being the largest dataset of its kind for pre-AML clonal hematopoiesis, is severely underpowered to fit this type of model. There is, in fact, barely enough statistical power even to reliably fit simple, non-additive, dominance models on the UK biobank, where there is rich phenotypic data and complete genotypes on 500,000 patients (see preprint by Pazokitoroudi et al. 2020). These models found no statistically significant evidence for non-additive effects in any of the 50 traits considered in the cited study. Note, that in our previous revision, we already state that our inability to model epistatic interactions is a limitation of our approach:

(Discussion, Page 17) Future work with long-read sequencing and longitudinal sampling, is required to experimentally phase somatic mutations genome-wide, and will help determine whether such mutations are occurring in the same or different clones thus allowing us to investigate the effects of both individual mutations as well as the epistatic interactions between linked mutations and clonal trajectories.

REVIEWERS' COMMENTS

Reviewer #5 (Remarks to the Author):

In this work the authors trained an ensemble of neural networks to predict different classes of mutations on synthetic simulation data sets. They then use the ensemble to analyze the mutational burden of real samples. I find the work interesting and the paper well written (besides some typos, and a few paragraphs that I find hard to understand). I believe this work has value and presents an interesting way of analyzing and assessing the complex impacts of mutations. I only have minor comments.

General remarks:

=====

From what I understood, the DNNs take summary statistics as inputs. These are numbered to 16, having 512 units in the hidden layers appears very excessive. Have the authors tried to train networks with smaller hidden layer sizes? I am not concerned about overfitting boosting test scores in this particular setting because the training has been made on simulated data and all important conclusions are drawn from real data. However, having that many parameters for a network with only 16 units as inputs and trained with adam can cause the model to overfit and decrease the overall generalization properties. I wonder if the accuracy of leaner networks would not be higher, especially for hard cases with little examples, like those for the "negative" vs "neutral" classes. How did the authors arrive and decide on 512 units?

Reproducibility:

=====

The code is made available and via github repository and it looks well organized.

Statistical tests:

=====

The tests used are valid.

Methods:

=====

1)

I appreciate that authors took the time to make the code available. It would be great if the authors could provide in the methods the exact description of the DNNs including:

- * All layer sizes
- * All non-linearities used
- * All losses used
- * Learning rate
- * Optimizer method (ex: adam)
- * Number of epochs
- * Regularizations
- * etc...

That would spare the reader the need to go look for them in the code.

A description of the method used for hyper-parameter optimization would also be appreciated (grid, manually optimized, Bayesian hyper-parameter optimization, ...).

Also a short indication of the time that it took for training and what hardware, would give the reader an idea of the kind of resources they would need for similar experiments.

II)

The DNNs trained are multi-objective however it is not clear from the manuscript what are the benefits of adding the objectives besides the class prediction. Could the authors provide an ablation test demonstrating that these additional costs improve class prediction, as well as measures of performances for the additional costs. I also suggest adding a few learning curves on different objectives in supplementary material. Multi-objective optimization can be tricky and these curves would give an idea on how stable the training is.

Figures:

=====

Figure1:

Suggestions:

- * State explicitly what goes into the input layer, in the same way as for the output layers.
- * Add the dimensionality of all layers (input, hidden, and output).

This would make it easier for the reader to follow the DNN part.

Figure 2b:

Could the authors add tests identifying which bins are significant when comparing control with preleukimian. This would indicate right away which evolutionary classes have been identified as the most significant differences between the two conditions.

Figure 2c:

"We show the estimated mutation rate for each sample from each DNN in our classifier (grey)". I find this sentence confusing. The DNNs have been referred to as "classifiers", here I assume that the ensemble is referred to as the "classifier". If that is the case another word such as "ensemble" would remove the confusion. The authors could also add to the legend that the samples are sorted according to the mean.

Figure 3a)

The authors should indicate which tests are significant on the graph.

Figure 3b)

I see only a blank white square on the left, is there something missing? If not could the other remove it?

Figure 4b)

The "b)" needs to be in bold.

Supp Fig1)

I suggest using a smaller size for the points and lowering the opacity. This would solve some of the occlusion problems (like the blob as (0, -30) when three classes are overlapping) and give an idea of the overall distributions.

Supp Fig2)

a, b are minuscule everywhere but in the legend of this figure.

Supp Fig7)

The authors should indicate the significance on the figure, and state what test was used in the legend.

General suggestion for figures:-----

Whenever a p-value is calculated the authors should also state the number of points used for calculating it.

Typos:

=====

The authors should proofread the text. These are the typos I found, there might be others. "selection" in: "In our simulation, we only selection to act on non synonymous"
"While we modelling probabilities" in: "Probability of a mutation being beneficial. While we modelling probabilities"

Wording:

=====

I find some paragraphs (such as the following) a bit hard to understand:"Using a linear regression, we compare the relationship between the number of mutations falling into known driver genes versus non-driver genes for cases and controls fitting the combination and positive evolutionary classes. Among participant sequences falling within the combination model, we found a significant interaction between the number of mutations falling in non-driver genes to driver for controls (B = 5.76) versus cases (B = 0.642); $F(1, 224) = 28.5$, $p = 2.23e-07$. However, in the positive class, we did not find this interaction to be significant (i.e., controls (B = 0.16) and cases (B = 0.17); $F(1, 12) = 0.0004$, $p = 0.98$)."

REVIEWERS' COMMENTS

Reviewer #5 (Remarks to the Author):

In this work the authors trained an ensemble of neural networks to predict different classes of mutations on synthetic simulation data sets. They then use the ensemble to analyze the mutational burden of real samples. I find the work interesting and the paper well written (besides some typos, and a few paragraphs that I find hard to understand). I believe this work has value and presents an interesting way of analyzing and assessing the complex impacts of mutations. I only have minor comments.

We thank the reviewer for their helpful feedback.

General remarks:

=====

From what I understood, the DNNs take summary statistics as inputs. These are numbered to 16, having 512 units in the hidden layers appears very excessive. Have the authors tried to train networks with smaller hidden layer sizes? I am not concerned about overfitting boosting test scores in this particular setting because the training has been made on simulated data and all important conclusions are drawn from real data. However, having that many parameters for a network with only 16 units as inputs and trained with adam can cause the model to overfit and decrease the overall generalization properties. I wonder if the accuracy of leaner networks would not be higher, especially for hard cases with little examples, like those for the "negative" vs "neutral" classes. How did the authors arrive and decide on 512 units?

We initially trained a classifier with smaller hidden layer sizes. However, as the reviewer anticipates below, there was a decrease in classification accuracy when we added the other outputs and trained a multi-objective neural network. Increasing the size of the hidden layer gives the network more capacity, and when we did so, our classification accuracy improved.

To avoid overfitting, we set regularization parameters and did early-stopping using a validation set. Our classifier is not overfit: it has similar classification accuracy on a very distinct test set as that on both our training and test set from the initial simulated data.

Reproducibility:

=====

The code is made available and via github repository and it looks well organized.

Statistical tests:

=====

The tests used are valid.

Methods:

=====

I) I appreciate that authors took the time to make the code available. It would be great if the authors could provide in the methods the exact description of the DNNs including:

- * All layer sizes
- * All non-linearities used
- * All losses used
- * Learning rate
- * Optimizer method (ex: adam)
- * Number of epochs
- * Regularizations
- * etc...

That would spare the reader the need to go look for them in the code.

A description of the method used for hyper-parameter optimization would also be appreciated (grid, manually optimized, Bayesian hyper-parameter optimization, ...).

Also a short indication of the time that it took for training and what hardware, would give the reader an idea of the kind of resources they would need for similar experiments.

These are good recommendations, thank-you. We have updated the Methods section in our manuscript to include the requested details.

II) The DNNs trained are multi-objective however it is not clear from the manuscript what are the benefits of adding the objectives besides the class prediction. Could the authors provide an ablation test demonstrating that these additional costs improve class prediction, as well as measures of performances for the additional costs. I also suggest adding a few learning curves on different objectives in supplementary material. Multi-objective optimization can be tricky and these curves would give an idea on how stable the training is.

Each network in the DNN ensemble has trained on a different cross-validation fold, so if our training was unstable, we would have expected to see highly variable outputs across the different DNNs in our ensemble. This is not what we observe. The error-bars in supplementary figure 2 show the standard error in max softmax probability across our ensemble for all individual patients. In general, the patients with less than 1.0 max probability are those where the classification is either negative or neutral selection. These two types of selection are famously hard to distinguish in population genetic analyses because, under many circumstances, negative (aka purifying) selection generates population statistics nearly identical to those from neutral selection with a smaller mutation rate. Indeed, we see this in the overlap in the distributions of the inputs associated with negative and neutral classes in the training data (Supplementary Figure 1).

Supplementary Figure 1. Distribution of summary statistics from simulated data and observed data. *t*-Distributed Stochastic Neighbor Embedding (*t*-SNE) mapping the distribution of 16 summary statistics for simulated and observed blood populations. Points were sampled from four different evolutionary classes (total sample size: $n = 20,000$) and are shown in colour (positive = green, negative = red, combination = orange, neutral = blue). Colour intensity was reduced to 70% to increase visibility. Observed summary statistics from the EPIC cohort are shown in black). Source data are provided as a Source Data file.

Supplementary Figure 2. Uncertainty associated with class predictions in EPIC cohort. A benefit of our ensemble-based approach is that, for each blood cell population, each DNN emits a softmax probability distribution across the four overarching evolutionary classes. In a conventional classification task, the class with the highest probability will be selected as the best fit. However, as we are employing an ensemble-based approach, we obtain a distribution of predictions for each population so as to measure the uncertainty associated with each prediction. To obtain the best fit evolutionary class for each individual, we calculated the mean and standard error for each softmax probability across the four evolutionary classes and accepted the class with the maximum softmax probability as the class of best fit. a) Here, we show the mean (dark blue) and standard error (light blue) softmax probability for all EPIC participants. We find that for approximately half of the participants ($n=215$ (51.5%)) we obtain average probability distributions of over 99%, with a maximum standard error of 0.008, indicating that we can predict overarching evolutionary classes with high certainty. b) All participants predicted with a high degree of certainty are classified as evolving under either beneficial or mixture models of evolution. Participants classified as evolving under negative or neutral classes of evolution typically exhibit increased levels of predictive uncertainty. Each boxplot illustrates the distribution of standard error associated with each prediction across evolutionary classes. The midline represents the medians, the upper and lower bounds the interquartile ranges, and the whiskers extend to 1.5 times the interquartile range. Boxplots are coloured according to evolutionary class (positive($n=28$): green, negative($n=63$): red, combination($n=228$): orange, neutral($n=158$): blue). Source data are provided as a Source Data file.

We are not adding additional outputs to improve the classification accuracy of the overarching evolutionary class. We are actually interested in what those values are; for example, Supplementary Figure 10, shows mutation rate estimates.

Supplementary Figure 10. Impact of population size on mutation rate. The expected number of mutations in a population (θ) is a product of the population size N and the per generation mutation rate (μ): $\theta=4N\mu$. Estimates of mutation rate can be extended to account for the range of population size estimates (10,000 – 200,000) which exist for the hematopoietic stem cell population. Here, we show the best fit per generation mutation rate on the (y axis) for each sample (x axis, $n = 477$). Samples are sorted according to mean mutation rate. Mutation rates have been scaled across varying HSC population size estimates. Source data are provided as a Source Data file.

The reason we don't make more use of these values is that we found that while we can distinguish between overarching evolutionary classes with high accuracy, as well as the presence or absence of positive or negative selection, our model struggles to discriminate amongst the weaker coefficients of selection which is notoriously challenging in population genetics.

We left the other outputs in our DNNs because we think that it is important to present both the successes as well as the anticipated challenges. However, we agree with the reviewer that this could be made more explicit and have included the following text:

[Page 7] Further, while we can distinguish between overarching evolutionary classes with high accuracy, as well as the presence or absence of positive or negative selection, our model struggles to discriminate amongst the weaker coefficients of selection which is notoriously challenging in population genetics (12). As such, we limit our inferences of selective dynamics in blood to the overarching evolutionary class which we are able to discriminate with a high accuracy.

Also, as the reviewer correctly anticipated, adding the extra outputs reduced the accuracy of the classification. However, as discussed above, expanding the size of the hidden layer, thereby increasing the capacity of the DNNs, re-established the high accuracy classification.

Figures:

=====

Figure1:

Suggestions:

- * State explicitly what goes into the input layer, in the same way as for the output layers.
- * Add the dimensionality of all layers (input, hidden, and output).

This would make it easier for the reader to follow the DNN part.

We have revised the figure to state what goes into the input layer and have included the dimensionality of all layers.

Figure 2b:

Could the authors add tests identifying which bins are significant when comparing control with preleukimian. This would indicate right away which evolutionary classes have been identified as the most significant differences between the two conditions.

We have updated the legend text in Figure 2b to indicate which bins are significant when comparing controls with preleukemic individuals.

Figure 2c:

"We show the estimated mutation rate for each sample from each DNN in our classifier (grey)". I find this sentence confusing. The DNNs have been referred to as "classifiers", here I assume that the ensemble is referred to as the "classifier". If that is the case another word such as "ensemble" would remove the confusion. The authors could also add to the legend that the samples are sorted according to the mean.

We have updated the legend for Figure 2c to replace the word “classifier” with “ensemble”. We have also revised the legend to state that samples are sorted according to the mean.

Figure 3a)

The authors should indicate which tests are significant on the graph.

We have updated Figure 3a to show which tests are significant on the graph.

Figure 3b)

I see only a blank white square on the left, is there something missing? If not could the other remove it?

Figure 3b refers to the upset plot in the lower panel. The white square in the upper left corner is due to the histogram (lower left corner) showing the number of genes per class and unfortunately cannot be removed. We have moved the labels to mitigate confusion.

Figure 4b)

The "b)" needs to be in bold.

We have removed all formatting (bold, italic etc.) to comply with the formatting requirements requested by the journal but we thank the reviewer for this comment.

Supp Fig1)

I suggest using a smaller size for the points and lowering the opacity. This would solve some of the occlusion problems (like the blob as (0, -30) when three classes are overlapping) and give an idea of the overall distributions.

We have updated the figure to lower the opacity and use a smaller size for the points.

Supp Fig2)

a, b are minuscule everywhere but in the legend of this figure.

We have updated the legend of this figure to correct the formatting of “a” and “b” in the legend of the figure to be consistent with the other figures.

Supp Fig7)

The authors should indicate the significance on the figure, and state what test was used in the legend.

We have revised Supplementary Figure 7 to indicate the significance on the figure and stated what test was used in the legend.

General suggestion for figures:-----

Whenever a p-value is calculated the authors should also state the number of points used for calculating it.

We have revised the text to include the number of points used to calculate p-values where not previously stated.

=====
Typos:

The authors should proofread the text. These are the typos I found, there might be others. "selection" in: "In our simulation, we only selection to act on non synonymous"

"While we modelling probabilities" in: "Probability of a mutation being beneficial. While we modelling probabilities"

We thank the reviewer for highlighting these errors and have corrected them in the text. We have proofread the text and made minor edits where necessary.

=====
Wording:

I find some paragraphs (such as the following) a bit hard to understand: "Using a linear regression, we compare the relationship between the number of mutations falling into known driver genes versus non-driver genes for cases and controls fitting the combination and positive evolutionary classes. Among participant sequences falling within the combination model, we found a significant interaction between the number of mutations falling in non-driver genes to driver for controls ($B = 5.76$) versus cases ($B = 0.642$); $F(1, 224) = 28.5$, $p = 2.23e-07$. However, in the positive class, we did not find this interaction to be significant (i.e., controls ($B = 0.16$) and cases ($B = 0.17$); $F(1, 12) = 0.0004$, $p = 0.98$)."

We thank the reviewer for this edit and have revised our language to increase clarity in this specific section. Additionally, we have reviewed the paper for clarity throughout.